# Algorithm for post-clustering curation of DNA amplicon data yields reliable biodiversity estimates

Tobias Guldberg Frøslev [1,2], Rasmus Kjøller[1], Hans Henrik Bruun [1], Rasmus Ejrnæs[3], Ane Kirstine Brunbjerg[3], Carlotta Pietroni[2] & Anders Johannes Hansen[2]

DNA metabarcoding is promising for cost-effective biodiversity monitoring, but reliable diversity estimates are difficult to achieve and validate. Here we present and validate a method, called LULU, for removing erroneous molecular operational taxonomic units (OTUs) from community data derived by high-throughput sequencing of amplified marker genes. LULU identifies errors by combining sequence similarity and co-occurrence patterns. To validate the LULU method, we use a unique data set of high quality survey data of vascular plants paired with plant *ITS2* metabarcoding data of DNA extracted from soil from 130 sites in Denmark spanning major environmental gradients. OTU tables are produced with several different OTU definition algorithms and subsequently curated with LULU, and validated against field survey data. LULU curation consistently improves α-diversity estimates and other biodiversity metrics, and does not require a sequence reference database; thus, it represents a promising method for reliable biodiversity estimation.

[1] Department of Biology, University of Copenhagen, Universitetsparken 15, DK-2100 Copenhagen, Denmark. [2] Natural History Museum of Denmark, University of Copenhagen, Øster Voldgade 5-7, DK-1350 Copenhagen, Denmark. [3] Department of Bioscience, University of Aarhus, Grenåvej 14, DK-8410 Rønde, Denmark. Correspondence and requests for materials should be addressed to T.G.F. (email: tobiasgf@bio.ku.dk) or to A.J.H. (email: ajhansen@snm.ku.dk)

Quantifying biodiversity is a key aim of ecological science, but for the majority of organisms, species detection and identification are so demanding and costly that assessment of multi-taxon biodiversity is generally intractable[1,2]. High-throughput sequencing (HTS) of genetic markers, which have already become standard in microbiology, is a promising tool for rapid, reproducible and thorough censuses of eukaryotic biodiversity in complex ecosystems[3–5]. However, it is poorly studied whether reliable eukaryotic α-diversity metrics can be achieved from such methods, possibly due to a shortage of comprehensive inventories with paired sets of thorough inventory data and DNA data.

Not only are there sampling issues with regard to environmental DNA, but PCR and sequencing processes also generate errors[6,7], which, together with intraspecific variation, result in sequence richness far beyond the 'true' richness of the sampled biotic community. Given the incomplete reference databases for most organism groups, such errors are not easily separated from true OTU's.

Molecular, ecological and biodiversity studies based on HTS have mainly been developed and applied to microorganisms (bacteria and fungi), for which true diversity generally is poorly known. Such studies have often estimated very high levels of α-diversity[4,8,9]. These high diversity estimates from HTS data have by some been taken as a first glance into a hitherto un-sampled 'rare biosphere'[9], by others as an argument for deleting the rarest OTUs at some arbitrary level. While many rare OTUs are beyond doubt real biological entities[10], an appreciable fraction of rare OTUs are likely errors from PCR and sequencing[11]. The proportion of erroneous singletons has been estimated to 38% on average[12], and studies of bacterial mock communities have revealed that standard bioinformatic approaches result in many spurious OTUs[13]. Bioinformatic error-reduction has mainly been focussed on selective removal of low-quality reads[7,11] and so-called chimeric sequences[14,15], recently implemented in model-based pipelines[13]. Other approaches have focused on laboratory measures to reduce the number of PCR and sequencing errors and the potential impact of these[16–18]. Although these advances have greatly improved data based on amplicon sequencing, erroneous OTUs remain a critical issue.

Most data processing algorithms cluster sequences into OTUs based on similarity (often set to around 97% depending on organism group and marker region) approximately corresponding to recognized average genetic species boundaries. To minimize the number of errors, the tail of the OTU rank-abundance distribution is then often discarded at some arbitrary level (usually only discarding singletons, but often more), assuming that a large proportion of infrequent OTUs represent errors[4,11,12,19]. However, in real ecosystems, genuinely infrequent species should be expected to make up the lion's share of total richness. Thus, current approaches are likely to retain only dominant species, which may be adequate for assessing composition and turnover, but much less so for α-diversity.

Aiming for improved diversity estimates and a taxonomic composition better aligned with 'reality', we developed a co-occurrence based post-clustering curation method, LULU. The LULU algorithm excludes artefactual OTUs without discarding rare, but real OTUs. The core mechanism is the identification and merging of 'daughter' OTUs with consistently co-occurring, sequence similar, but more abundant 'parent' OTUs across a multi-sample data set, under the assumption that the 'daughter' OTUs are artefacts. The algorithm is independent of a reference database, and can thus be applied to any OTU table produced for any set of samples produced by any initial OTU definition algorithm. A related approach—distribution based clustering—

was developed to cluster 16S bacterial sequence data into ecologically significant OTUs[20], and recently implemented in the dbotu3 tool[21]. However, whereas LULU is a post-clustering curation method aiming at removing erroneous OTUs to achieve meaningful diversity metrics, dbotu3 is a clustering method aiming at identifying ecologically significant haplotypes of bacterial strains. Despite different objectives and parameters, the overall processing strategy of the dbotu3 algorithm is similar to LULU.

To validate the LULU algorithm, we used a plant data set for ITS2 (nuclear ribosomal internal transcribed spacer region 2) obtained from an extensive soil sampling across 130 field sites in Denmark, for which thorough reference data (presence/absence) of vascular plants were obtained concurrently. A total of 564 plant species (approximately one third of the naturally occurring flora) were recorded in the study, with field site species richness per site ranging from 6 to 93. For the main part of the validation, we used the botanical survey data as ground truth data for species-level α-diversity and composition. For additional parts of the validation procedure, we assigned OTUs to species in public reference databases (GenBank), which contained ITS2 data for 88% of species recorded in the field survey, allowing satisfactory taxonomic resolution for most OTUs.

OTU tables were produced from the sequence data using a set of representative clustering methods: VSEARCH[22], SWARM[23], CROP[24] and DADA2[13]. All tables were subsequently curated with the LULU algorithm, and the curation effect was evaluated against ground truth in the form of plant survey data. We also tested the dbotu3 algorithm, both for 'one-step' clustering as intended, and as an alternative to LULU for post-clustering curation. Furthermore, we evaluated the effect of singleton removal in comparison to curation. Finally, we visualized and evaluated the exact curation effect on selected genera of plants. We show that co-occurrence based post-clustering curation greatly improve diversity measures for all tested OTU tables for a large set of metrics. We conclude that LULU is a tool with far-reaching potential for practical application where realistic biodiversity metrics are needed.

## Results

**Curation improves correlation with plant richness.** OTU richness data for the un-curated and curated tables across the 130 sampling sites were regressed against the corresponding observed plant richness, for each clustering approach and similarity level separately (Fig. 1a (97% clustering level), Table 1, Supplementary Figs 1, 2 (all clustering levels)). Three measures for correspondence with 'real richness' were examined: (i) The coefficient of determination ($R^2$) was used as a measure of goodness of prediction, (ii) an intercept close to zero was expected in combination with (iii) a regression slope close to —but less than—unity to be indicative of a realistic prediction, reflecting systematic soil sampling to be less effective than a thorough botanical survey, although DNA from soil for several reasons may in fact contain DNA from more species than are apparent at the time of investigation (Methods section).

The application of the LULU algorithm consistently improved all selected measures of correspondence with survey data for all initial clustering approaches. The coefficient of determination ($R^2$) was improved by 0.03–0.49 (mean improvement 0.27). The two greedy algorithms, VSEARCH and SWARM, initially resulted in relatively poor fits with low $R^2$ values, intercepts well above zero and general overestimation of richness, whereas the two model-based algorithms, CROP and DADA2 (+VSEARCH), performed considerably better.

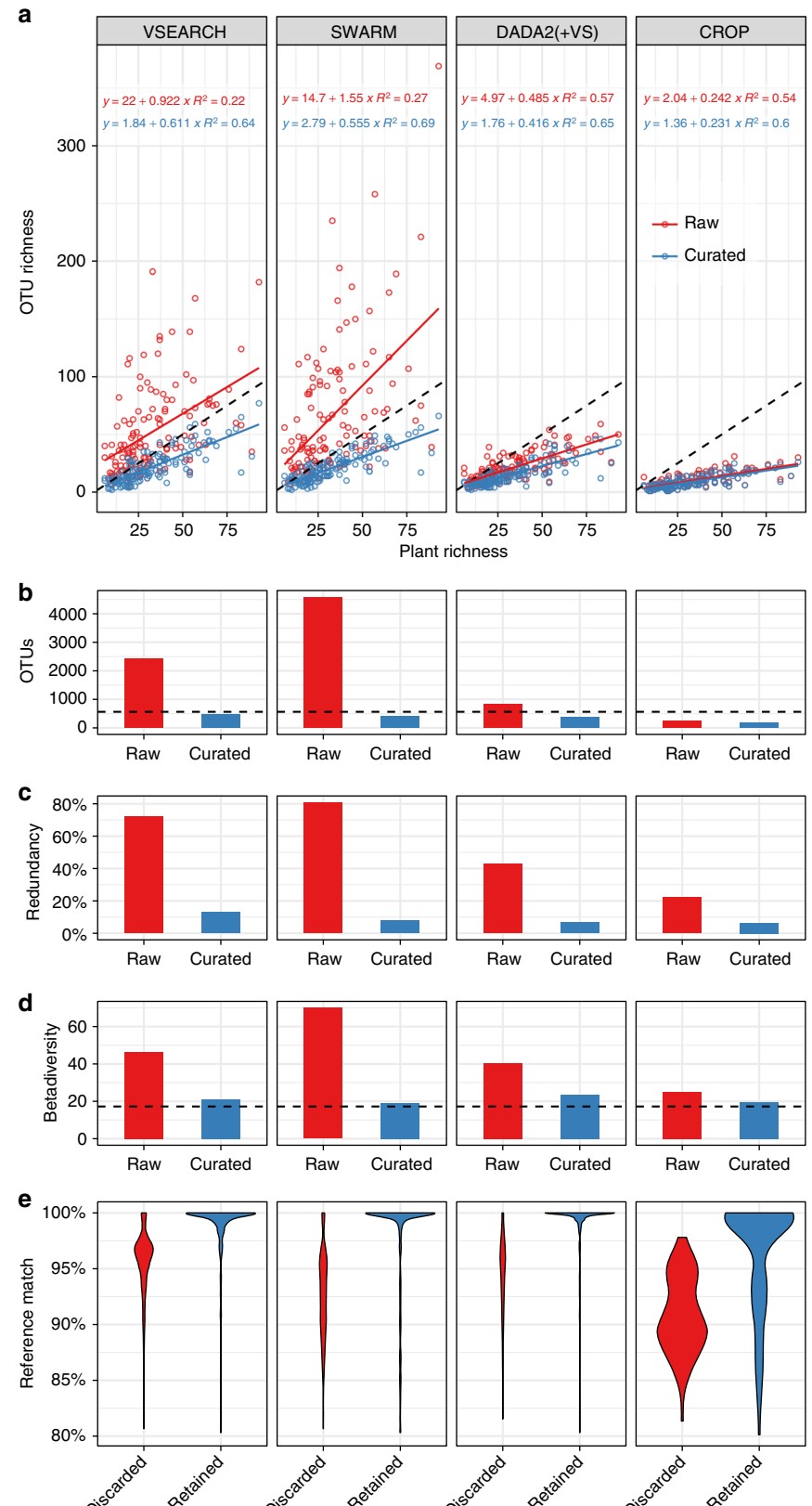

**Fig. 1** Effects of curation with the LULU algorithm for clustering methods at 97% level. OTU table metrics before (red = raw) and after (blue = curated) curation with LULU. **a** correspondence of OTU (plant *ITS2* sequence data) richness vs. plant richness for each of the 130 sampling sites, **b** total number of OTUs compared to total plant species recorded (564 species, dashed line), **c** percentage of OTUs having taxonomically redundant annotation, **d** OTU β-diversity (total richness/mean site richness) compared to plant β-diversity (17.23, dashed line), **e** distribution of best reference database (GenBank) match for OTUs retained and discarded by LULU

**Table 1 Metrics of the OTU tables produced with multiple OTU generation algorithms before and after curation with LULU**

| Method | Level | Correlation (R²) | Slope | Intercept | Taxonomic redundancy | Total OTUs | Avg. best match | β-diversity |
|---|---|---|---|---|---|---|---|---|
| CROP | 98% | 0.56/0.59 | 0.32/0.3 | 3.8/2.9 | 28%/7% | 369/241 | 95.8%/97.5% | 25.9/19.1 |
| CROP | 97% | 0.54/0.6 | 0.24/0.23 | 2/1.4 | 22%/6% | 249/174 | 94.7%/96.4% | 25/19.5 |
| CROP | 95% | 0.48/0.6 | 0.24/0.22 | 1.8/1.1 | 28%/8% | 383/252 | 92.2%/93.7% | 39/29.9 |
| DADA2 | 100% | 0.42/0.56 | 0.77/0.53 | 15.6/3.6 | 77%/45% | 2568/761 | 97.7%/98.8% | 62.8/36.3 |
| DADA2( + VS) | 98.50% | 0.54/0.63 | 0.55/0.44 | 6.4/1.8 | 53%/13% | 1141/430 | 96.7%/98.7% | 46.9/26.5 |
| DADA2( + VS) | 98% | 0.55/0.64 | 0.52/0.42 | 6/1.9 | 50%/10% | 1033/402 | 96.6%/98.7% | 45.2/25.5 |
| DADA2( + VS) | 97% | 0.57/0.65 | 0.49/0.42 | 5/1.8 | 43%/7% | 842/365 | 96.4%/98.6% | 40.4/23.7 |
| DADA2( + VS) | 96% | 0.62/0.67 | 0.47/0.41 | 4/1.3 | 37%/6% | 721/341 | 96.2%/98.6% | 37.3/22.9 |
| DADA2( + VS) | 95% | 0.61/0.68 | 0.44/0.41 | 3.7/1.1 | 32%/5% | 622/324 | 96.2%/98.5% | 34.2/22.3 |
| SWARM | 99% | 0.15/0.64 | 3.49/0.64 | 49.6/2.1 | 93%/18% | 14828/520 | 95.1%/97.9% | 90.5/22.5 |
| SWARM | 98.50% | 0.2/0.67 | 2.35/0.62 | 26.4/1.8 | 88%/13% | 8422/467 | 94.2%/97.8% | 81.5/21.2 |
| SWARM | 98% | 0.25/0.69 | 1.81/0.58 | 18.1/2.1 | 84%/9% | 5779/430 | 93.6%/97.7% | 74.8/20.6 |
| SWARM | 97% | 0.27/0.69 | 1.55/0.56 | 14.7/2.8 | 81%/8% | 4585/401 | 93.3%/97.7% | 70/19.1 |
| SWARM | 96% | 0.27/0.7 | 1.55/0.56 | 14.1/2.8 | 81%/8% | 4547/401 | 93.2%/97.7% | 70/19.1 |
| SWARM | 95% | 0.39/0.71 | 1.15/0.53 | 4.5/2.3 | 70%/9% | 2500/362 | 92.6%/97.3% | 59.4/18.5 |
| VSEARCH | 98.50% | 0.15/0.63 | 2.15/0.73 | 62.7/1.6 | 90%/23% | 8008/558 | 97.4%/98.4% | 60.2/21.9 |
| VSEARCH | 98% | 0.17/0.59 | 1.58/0.7 | 41.5/1.7 | 85%/20% | 4815/517 | 96.8%/98.4% | 51.6/20.9 |
| VSEARCH | 97% | 0.22/0.64 | 0.92/0.61 | 22/1.8 | 72%/13% | 2425/458 | 96.1%/98.4% | 46.5/21 |
| VSEARCH | 96% | 0.27/0.64 | 0.8/0.57 | 16.4/1.9 | 64%/10% | 1740/415 | 95.7%/98.3% | 40.9/20.1 |
| VSEARCH | 95% | 0.34/0.66 | 0.7/0.55 | 12.3/1.9 | 56%/9% | 1320/396 | 95.5%/98.2% | 37.5/19.8 |

Effects of post-clustering curation with the LULU algorithm for clustering methods (VSEARCH, SWARM, DADA2 and CROP) at several levels. Values before the slash represent metrics for the method prior to curation with LULU. Values after the slash are post-curation metrics. $R^2$ denotes the coefficient of determination of the linear regression of OTU count vs. plant richness, slope and intercept denotes the constants of the inferred linear regression, taxonomic redundancy is calculated as the proportion of OTUs with a redundant taxonomic assignment, total OTUs is the count of total unique OTUs for each method, avg. best match is the average of the best GenBank match for all OTUs for each method, and β-diversity is the average α-diversity divided by γ-diversity

**Total OTU richness compared to total survey species richness**. We further evaluated OTU definition and LULU curation from the total number of OTUs identified and retained (Fig. 1b (97% clustering level), Table 1, Supplementary Figs. 3 and 4 (all clustering levels)). For a realistic OTU definition, we expected a total OTU number not surpassing the actual number of plant species recorded across all sites (564 species in total). Again, VSEARCH and SWARM initially performed relatively poor by identifying 2.3–27 times more OTUs (1320–14,828) than observed plant species at all clustering levels. The DADA2 (+VSEARCH) reached much lower OTU numbers, even at low clustering levels, but still with an overestimation at all levels. The CROP algorithm was the only method to underestimate the total richness, but surprisingly the number of OTUs was not correlated with clustering level, showing the lowest OTU count at 97%. The application of the LULU algorithm consistently reduced the total number of OTUs to less than the maximum criterion of 564 for all approaches.

**Taxonomic redundancy**. The OTU definition and LULU curation were further evaluated by estimating the species-level taxonomic redundancy of each table (Fig. 1c (97% clustering level), Table 1, Supplementary Fig. 5 (all clustering levels)). GenBank is relatively well populated (88.8% coverage) with sequences assigned to species observed in our study, so we expected an ideal OTU definition to result in a low number of OTUs receiving a redundant taxonomic annotation (proportion of OTUs with a taxonomic annotation already represented by another OTU in the table). Once more, VSEARCH and SWARM initially performed relatively poor by having the highest levels of taxonomic redundancy (ranging from 56 to 93%). In comparison, the DADA2 (+VSEARCH) approach had lower redundancy at all levels. Surprisingly, the CROP algorithm, which retained the far lowest number of OTUs, still showed a high taxonomic redundancy (22–28%). Curation with LULU resulted in a marked reduction of taxonomic redundancy at all levels. After curation, taxonomic redundancy ranged from 6 to

13% at the 97% clustering level, as opposed to 22–81% before curation.

**β-diversity**. Many errors in amplicon based diversity studies can be assumed to be unique because they arise independently during PCR or sequencing and, thus, are sample specific. Therefore, data sets with such errors are expected to show higher β-diversity (compositional differentiation across samples), than ideal error-free data sets. Hence, a realistic OTU definition was assumed to produce an OTU β-diversity close to the β-diversity of the botanical survey (17.23). A simple β-diversity measure (average α-diversity divided by γ-diversity) was applied to all uncurated and LULU curated tables (Fig. 1d (97% clustering level), Table 1, Supplementary Fig. 6 (all clustering levels)). All initial clustering methods produced OTU β-diversity levels (25–90.5) exceeding field survey β-diversity at all clustering levels, again with VSEARCH and SWARM exhibiting the highest levels. For all clustering methods, curation with LULU resulted in β-diversity levels much closer to the botanical field survey than un-curated data, ranging from 19.1 to 26.5 in all approaches except the pure DADA2 approach with a value of 36.3.

**Distribution of reference database matches**. We expected an ideal curation algorithm to primarily retain OTUs with a perfect or near-perfect match to the reference database. We used two metrics, (i) the distribution of best matches and (ii) the average best match of all OTUs retained and discarded by LULU. We found a marked increase in average best match by LULU curation across clustering methods (Fig. 1e (97% clustering level), Table 1, Supplementary Fig. 7 (all clustering levels)). Curation by LULU consistently showed an improved distribution of best matches and also an increase in average match for all methods.

**Taxonomic composition**. To assess if LULU curation retained the 'correct' OTUs, and not only improved richness estimates and method-level metrics, we compared the taxonomic composition of OTUs with the list of species recorded for each site in the

**Table 2 Taxonomic composition of OTUs for single sites compared with plant survey data**

| Method | Level | Imperfect_matches | Recaptured species | Unregistered species | Redundant species | Lost species |
|---|---|---|---|---|---|---|
| CROP | 98% | 0.56 ± 0.16/0.50 ± 0.17 | 0.31 ± 0.13/0.34 ± 0.13 | 0.12 ± 0.11/0.13 ± 0.12 | 0.02 ± 0.05/0.02 ± 0.06 | 0.00 ± 0.01 |
| CROP | 97% | 0.80 ± 0.13/0.79 ± 0.13 | 0.13 ± 0.11/0.14 ± 0.11 | 0.06 ± 0.10/0.07 ± 0.10 | 0.00 ± 0.00/0.00 ± 0.00 | 0.00 ± 0.00 |
| CROP | 95% | 0.87 ± 0.09/0.86 ± 0.09 | 0.09 ± 0.09/0.10 ± 0.09 | 0.04 ± 0.09/0.04 ± 0.09 | 0.00 ± 0.00/0.00 ± 0.00 | 0.00 ± 0.00 |
| DADA2 | 100% | 0.66 ± 0.12/0.40 ± 0.14 | 0.22 ± 0.10/0.43 ± 0.16 | 0.08 ± 0.07/0.14 ± 0.11 | 0.03 ± 0.03/0.03 ± 0.05 | 0.02 ± 0.04 |
| DADA2( + VS) | 98.50% | 0.51 ± 0.15/0.29 ± 0.14 | 0.36 ± 0.14/0.54 ± 0.17 | 0.11 ± 0.08/0.16 ± 0.12 | 0.01 ± 0.03/0.02 ± 0.05 | 0.02 ± 0.09 |
| DADA2( + VS) | 98% | 0.48 ± 0.16/0.29 ± 0.14 | 0.38 ± 0.15/0.54 ± 0.15 | 0.12 ± 0.09/0.16 ± 0.12 | 0.01 ± 0.03/0.01 ± 0.03 | 0.02 ± 0.09 |
| DADA2( + VS) | 97% | 0.46 ± 0.16/0.29 ± 0.14 | 0.40 ± 0.14/0.53 ± 0.15 | 0.13 ± 0.09/0.16 ± 0.12 | 0.01 ± 0.03/0.02 ± 0.04 | 0.01 ± 0.03 |
| DADA2( + VS) | 96% | 0.42 ± 0.16/0.27 ± 0.14 | 0.43 ± 0.15/0.55 ± 0.16 | 0.13 ± 0.1/0.16 ± 0.12 | 0.01 ± 0.02/0.01 ± 0.03 | 0.01 ± 0.03 |
| DADA2( + VS) | 95% | 0.39 ± 0.16/0.25 ± 0.14 | 0.45 ± 0.17/0.56 ± 0.17 | 0.15 ± 0.1/0.19 ± 0.12 | 0.00 ± 0.02/0.00 ± 0.02 | 0.01 ± 0.03 |
| SWARM | 99% | 0.80 ± 0.15/0.32 ± 0.12 | 0.11 ± 0.10/0.46 ± 0.14 | 0.05 ± 0.05/0.19 ± 0.10 | 0.03 ± 0.03/0.03 ± 0.05 | 0.02 ± 0.05 |
| SWARM | 98.50% | 0.74 ± 0.16/0.29 ± 0.11 | 0.15 ± 0.11/0.48 ± 0.13 | 0.08 ± 0.07/0.22 ± 0.10 | 0.03 ± 0.04/0.01 ± 0.02 | 0.02 ± 0.05 |
| SWARM | 98% | 0.69 ± 0.17/0.26 ± 0.12 | 0.18 ± 0.11/0.49 ± 0.14 | 0.10 ± 0.08/0.24 ± 0.11 | 0.03 ± 0.03/0.01 ± 0.03 | 0.03 ± 0.05 |
| SWARM | 97% | 0.66 ± 0.17/0.25 ± 0.12 | 0.20 ± 0.11/0.48 ± 0.14 | 0.12 ± 0.09/0.27 ± 0.11 | 0.02 ± 0.03/0.00 ± 0.01 | 0.03 ± 0.05 |
| SWARM | 96% | 0.65 ± 0.17/0.25 ± 0.12 | 0.20 ± 0.11/0.49 ± 0.14 | 0.13 ± 0.09/0.27 ± 0.11 | 0.02 ± 0.03/0.00 ± 0.01 | 0.03 ± 0.05 |
| SWARM | 95% | 0.55 ± 0.17/0.24 ± 0.11 | 0.27 ± 0.13/0.48 ± 0.16 | 0.16 ± 0.10/0.28 ± 0.13 | 0.02 ± 0.04/0.00 ± 0.01 | 0.05 ± 0.08 |
| VSEARCH | 98.50% | 0.85 ± 0.09/0.43 ± 0.14 | 0.10 ± 0.07/0.42 ± 0.15 | 0.04 ± 0.03/0.14 ± 0.09 | 0.01 ± 0.02/0.01 ± 0.04 | 0.02 ± 0.05 |
| VSEARCH | 98% | 0.80 ± 0.12/0.41 ± 0.16 | 0.13 ± 0.09/0.44 ± 0.16 | 0.05 ± 0.05/0.15 ± 0.09 | 0.02 ± 0.02/0.01 ± 0.03 | 0.02 ± 0.05 |
| VSEARCH | 97% | 0.70 ± 0.14/0.39 ± 0.15 | 0.21 ± 0.11/0.45 ± 0.14 | 0.08 ± 0.06/0.15 ± 0.10 | 0.02 ± 0.03/0.01 ± 0.03 | 0.02 ± 0.04 |
| VSEARCH | 96% | 0.64 ± 0.14/0.36 ± 0.14 | 0.25 ± 0.11/0.47 ± 0.14 | 0.09 ± 0.07/0.16 ± 0.10 | 0.02 ± 0.03/0.01 ± 0.03 | 0.01 ± 0.03 |
| VSEARCH | 95% | 0.60 ± 0.14/0.36 ± 0.14 | 0.28 ± 0.11/0.47 ± 0.15 | 0.10 ± 0.07/0.16 ± 0.10 | 0.02 ± 0.03/0.00 ± 0.01 | 0.01 ± 0.03 |

Effect of curation on the taxonomic composition of single sites for OTU tables produced with different clustering methods at several levels. Values before the slash are values prior to curation with LULU. Values after the slash are post-curation values. Values are average proportions for single sites (given with standard deviations). Imperfect matches are calculated as the proportion of OTUs for each site that have a less than 100% reference database match. Recaptured species are calculated as the proportion of OTUs with a perfect reference database match and a unique taxonomic annotation corresponding to a plant species recorded for the site. Unregistered species are calculated as the proportion of OTUs with a perfect reference database match and a unique taxonomic annotation corresponding to a plant species not recorded for the site. Redundant species are calculated as the proportion of OTUs with a perfect reference database match and a redundant taxonomic annotation (i.e., already represented by a recaptured or unregistered species). Lost species is the proportion of the recaptured species lost during curation

survey (Table 2). For all methods, LULU curation resulted in smaller proportion of OTUs with imperfect reference database matches (mean of 0.64 before curation, mean of 0.38 after curation) corroborating the method-level results (see above), i.e., that curation mainly discarded imperfect matches, which are likely errors. Likewise, the proportion of recaptured species (i.e. OTUs with a perfect database match and a unique taxonomic annotation corresponding to a plant species recorded in the survey) increased by curation for all clustering methods (mean of 0.24 before curation, mean of 0.44 after curation). The proportion of unregistered species (i.e. OTUs with a perfect match and a unique taxonomic annotation corresponding to a species not recorded in the survey) also increased for most methods (mean of 0.10 before curation, mean of 0.17 after curation). CROP showed a higher level of imperfectly matching OTUs and results were not improved much by curation—confirming the method-level indications, that CROP selects suboptimal representative sequences (with lower reference database matches). For most methods, LULU curation resulted in a small proportion (mean 0.02) of the initially recaptured species being lost again, i.e. discarding true species occurrences.

**Community dissimilarity**. Metrics of community composition is mainly driven by the dominant and widespread species, and we hypothesized (i) that curation would have no major impact on dissimilarity measures based on uncurated vs curated OTU tables, and (ii) that a valid curation could not make the correlation between dissimilarity measures based of survey data and OTU data larger by curation. To test these hypotheses, we estimated community dissimilarity of all 40 OTU tables and the plant survey data with the Bray-Curtis metric. For plant survey data the dissimilarity metrics were calculated only for binary (presence/absence) data, but for sequence data we also tested metrics based on abundance (read count) data, as this in common practice, and potentially could yield stronger correlations with observational data.

Mantel tests (Supplementary Table 1) showed that all pairs of dissimilarity matrices (from un-curated vs. curated OTU tables) were highly correlated, with mantel r-statistics between 0.761 and 0.993 (all p-values < 0.001) when based on binary data, and

between 0.987 and 1 when based on abundance data. The lowest r-statistics were observed for dissimilarity matrices based on binary versions of OTU tables from greedy algorithms at lower clustering levels. These tables had the highest number of low-abundance OTUs removed by curation, and thus the effect on dissimilarity metric based on binary data is larger.

Comparing dissimilarity matrices for OTU tables with the dissimilarity matrix for plant data (Supplementary Table 2), revealed that all were highly correlated, with mantel r-statistics of 0.57–0.78 (avg. 0.68, all p-values < 0.001). Correlations were higher, when dissimilarity was evaluated for binary data (r-values of 0.63–0.76, avg. 0.70, and r-values of 0.67–0.78, avg. 0.751 after curation) and lowest when including information on read abundances (r-values of 0.57–0.66, avg. 0.63, and r-values of 0.57–0.67, avg. 0.64 after curation). For all 20 OTU tables, curation resulted in dissimilarity matrices with the same or slightly improved correlation with plant data, with largest improvement of binary data and OTU tables from the greedy algorithms using lower clustering.

**Singleton removal compared to post-clustering curation**. To compare the traditional noise-removal approach of singleton removal to post-clustering curation, we removed singletons (observations with a read count of one) from the initial OTU tables and compared the resulting metrics with those of the corresponding LULU curated tables (including singletons) (Supplementary Figs 8–15, Supplementary Tables 3 and 4). Singleton removal had some positive impact on several measures, especially for the approaches using greedy algorithms and low clustering levels. But no metrics were improved to a degree similar to LULU curation, e.g. the coefficients of determination ($R^2$) for the correlation between OTU richness and plant richness showed a mean improvement of 0.03 by singleton removal, compared to the mean improvement of 0.27 with LULU curation. DADA2 retains very few singletons during the processing (eight in this study), but the metrics of the DADA2 approaches were still superior to the other algorithms after removal of their singletons.

**Post-clustering curation with dbotu3 compared to LULU**. Although intended as a one-step clustering algorithm, we tested

whether the dbotu3 algorithm could be used for post-clustering curation as an alternative tool to LULU. We applied the method with two different settings (i) an abundance criterion of 0 to account for only sequencing errors, and (ii) an abundance criterion of 10 to merge ecological populations. We compared the results of this alternative curation with the results of LULU for most of the same basic metrics used in the validation of LULU (Supplementary Figs 8–15, Supplementary Tables 3 and 4). As expected due to the related structure of the algorithm, dbotu3 could be applied as a post-clustering curation. The application of dbotu3 to the clustered OTU tables resulted in highly curated tables with improved metrics for all investigated measures for both settings. The most pronounced curation effect was achieved with the approach aimed at merging ecological population (abundance criterion 0), and the effect came close to that of LULU for most measures. Nevertheless LULU performed better in all comparisons except a few metrics (avg. best match for the CROP tables), especially when applied to the OTU tables produced with the greedy algorithms, SWARM and VSEARCH.

**Distribution based OTU clustering**. We also applied the distribution based clustering algorithm (dbotu3), as a one-step clustering approach—as intended for this method. We applied it to our data in the form of an OTU table of unclustered reads, and compared the results with our plant survey data and with the other clustering approaches. We applied the method with the same two settings as above, and compared the resulting tables to our plant survey data (Supplementary Fig. 16, Supplementary Tables 2–5). For both settings, the reference database match was the highest observed in this study, and beta diversity was the lowest, lower than for the plant survey data, while the community dissimilarity metrics (comparison to plant based community dissimilarity) were comparable to that of the other initial clustering algorithms. In all other regards, there was little correlation with plant data. The second approach ($a = 0$, merging of ecological population) had slightly better metrics but the general performance of dbotu3 was comparable to that of the two greedy algorithms (VSEARCH and SWARM) without post-clustering curation. The processing time with dbotu3 was by far the longest of the processes applied in this study, being 17 days and 10.5 days, respectively, for the two settings.

**Curation effect on selected plant genera**. We evaluated the more detailed effects of curation for selected genera of plants. We plotted the abundance (read count) and best match of all OTUs assigned to a genus and compared the curation effect against occurrence data from the plant survey (Supplementary Figs. 17–32). For individual genera, the effect of curation confirmed the overall results as presented above: more realistic richness estimates, lowered taxonomic redundancy and better match with reference data. The number of taxonomically redundant OTUs varied considerably with clustering algorithm and clustering level but, for all genera, curation had a marked homogenizing effect across methods. The plant genera *Fagus* and *Calluna* are examples of genera with only one species represented in the study area, and that single species being abundant in field survey and sequencing data. For such genera, it was evident that the number erroneous (or at least taxonomically redundant) OTUs was high for most methods, and that LULU curation resulted in realistic levels of diversity. For *Fagus*, it was interesting to note that a single OTU with a 100% match and impervious to curation, was in fact a fungal sequence wrongly annotated as *Fagus* in GenBank. As the reads of this fungal OTU had distribution and abundance patterns contrasting that of *Fagus*, it was

not discarded by the LULU algorithm. Many errors were highly abundant and would not be excluded by a universal abundance cutoff without simultaneously removing a large number of real, but low-abundant species. With clustering levels from 98% and downwards (97%, 96%, and 95%), the rate of undesired clustering of real species seemed to increase rapidly. However, the most inclusive clustering level (95%) still retained redundant/erroneous OTUs. In many instances, the CROP algorithm initially identified a few OTUs correlating with results from the other approaches, including LULU curation. However, CROP entirely missed the OTUs identified by other algorithms or selected suboptimal representative sequences with lower reference database matches.

## Discussion

We developed and validated a post-clustering algorithm, LULU, with the aim to retain true α-diversity and taxonomic composition, while discarding the artefactual OTUs from community data derived by HTS of marker genes (metabarcoding). We showed that LULU significantly improved the a-diversity signatures when applied to a range of different OTU assignment tools. Although LULU was validated for vascular plants the method is particularly attractive for organism groups with poorly populated reference databases and for which traditional ways to estimate α-diversity are tedious (e.g., many groups of invertebrates) or impossible (e.g., protozoa, fungi, and bacteria).

In contrast to ecological studies, where increased sampling effort leads to more reliable diversity estimates, the proportion of erroneous singletons in HTS studies is expected to approach 100% asymptotically with increased sequencing depth, even with low (1%) error rates[25]. Apart from the direct diversity inflation, this further complicates the application of extrapolating richness estimators, which generally depend on the number of singletons to estimate the number of un-recorded taxa[4], and it has been recommended that richness estimation should be avoided all together for HTS data[4]. To obtain more realistic data, it has been advocated e.g. to remove singletons[4], OTUs occurring in abundances of 5–10 or below[12], or to resample to even sequencing depth[26], whereas others find it inadmissible to delete valid data[27]. None of these approaches aim to identify the actual artificial taxa. In this study, we show that the post-clustering curation with LULU successfully identifies and removes a large proportion of the remaining erroneous OTUs without the need of applying arbitrary cutoff levels. Furthermore, our additional measures of validity (i.e., total OTUs richness, taxonomic redundancy, β-diversity, reference database matches, taxonomic composition, and community composition) indicate an improved OTU definition for a wide range of purposes (e.g., comparison with other studies, correspondence with reference databases, taxonomic composition, and general community ecological studies). Traditional removal of singletons had a positive effect on several measures, but remained much less effective than LULU curation. Although curation with LULU improved community dissimilarity metrics, our results confirmed that existing approaches to OTU tables are adequate for studies of community dissimilarity. The 'cost of curation' is a small loss of 'real species'. The lost species will most likely be rare, low-abundance and low-occurrence OTUs co-occurring with closely related more abundant species (i.e., species having a distribution like that of errors), and LULU curation may thus not be suitable for identifying single rare species in community data sets. We have developed and tested the algorithm on sequence data obtained from environmental samples aimed for actual biodiversity studies. Although, mock communities never will have the complexity of

real life soil samples, future validation of LULU and related approaches on mock communities with known genetic content will be valuable.

The improvement by LULU curation was most pronounced for the two greedy methods, VSEARCH and SWARM, which initially had the highest number of OTUs. At a clustering level usually considered biologically realistic (97%), these methods overestimated true richness at least four-fold (2425 and 4585), but were successfully curated for realistic measures with LULU. The initial metrics for the model-based approaches, CROP and DADA2, were more realistic, but were still appreciably improved by curation. CROP was the method least receptive to curation—mainly because the initial number of OTUs was much lower. Despite a relatively good initial prediction of α-diversity, the OTU picking of CROP seemed to be suboptimal compared to the other methods investigated, as the similarity between OTU and best GenBank match (Fig. 1e) were far lower, and the taxonomic redundancy (Fig. 1c) and β-diversity (Fig. 1d) measures performed poorly considering the markedly lower number of OTUs (Fig. 1b). Although DADA2 probably had the highest proportion of true sequences, the biological co-occurrence patterns of these obstructed further curation with LULU towards reliable species-level richness (Supplementary Figs 1–7, Table 1). However, clustering of the initial OTUs enabled using the efficient DADA2 algorithm for OTU definition and subsequent curation with LULU. Future incorporation of LULU or LULU-like algorithms in pipelines like DADA2 may be promising.

LULU outperformed dbotu3 as a post-clustering curation tool. This was most pronounced for the OTU tables produced with the greedy algorithms VSEARCH and SWARM—i.e., the tables with a high proportion of erroneous OTUs. The reason for this difference may be that dbotu3 uses the Levenshtein edit distance for pairwise sequence comparison, whereas LULU as applied here uses the BLASTn algorithm, which may better identify erroneous sequences with larger gaps and insertions, not initially absorbed by the greedy clustering algorithms. As a stand-alone clustering algorithm (as originally intended), dbotu3 performed worse than any of the other algorithms in combination with post-clustering curation by LULU.

LULU seemed to have a homogenizing effect on the widely different initial clustering results, in the sense that all the methods resulted in relatively good predictions of α-diversity and much more similar and improved results for the other diversity metrics after curation with LULU (Table 1, Supplementary Figs 1–7). On the basis of our study we recommend an approach applying initial OTU definition with DADA2, subsequent clustering (with, e.g., VSEARCH) addressing the average intraspecific variation of the target group, and curation with LULU as a safe pathway to obtain reliable and accurate data, without discarding much true information.

In conclusion, the post-clustering algorithm LULU can greatly improve the accuracy of richness estimates derived from amplicon sequencing. This is achieved by excluding erroneous OTUs and thereby reducing taxonomic redundancy and improving similarity with true community composition. Here, we applied LULU to 20 different initial OTU definition approaches, and found that all metrics of correspondence, taxonomic redundancy and composition and community composition were improved. Given the normal levels of plant community species richness and of intraspecific genetic variability in plants, we believe our method validation is relevant to other organism groups and markers less easy to evaluate due to the lack of reference survey data. LULU is independent of a reference database and applicable to all types of amplicon data from studies with a series of samples. Considering the rapidly growing interest in metabarcoding for biodiversity assessment, LULU is a tool with far-reaching

potential for practical application, as it is an efficient tool to exactly assess OTU-based diversity as long as error-free sequencing and all-inclusive databases still are to be developed.

## Methods

**Assumptions and objectives of the LULU algorithm.** The algorithm is intended as a post-clustering OTU table curation method aimed at removing erroneous OTUs from tables produced by any clustering algorithm e.g., methods used in this study[13,22–24], and those implemented in Qiime[28] and Mothur[29], as long as the product is an OTU table and a corresponding file with representative sequences.

The implementation of the algorithm is based on a set of assumptions based on four observations we have previously made when working with HTS of amplified marker genes (a.k.a. metabarcoding) of well-studied organism groups with well-populated reference databases present (i.e., plants, as used for validation here). The first observation is that OTU tables often have more OTUs than expected from biological knowledge of the system under investigation[11]. The second observation is that OTU tables often contain low-abundance OTUs, which are taxonomically redundant in the sense that their taxonomic assignment is identical to more abundant OTUs. This pattern may be caused by incomplete reference data and/or insufficient clustering, but can also indicate that the OTU is effectively a methodological artefact. The third observation is that the highest sequence similarity (match rate) of such taxonomically redundant, low-abundance OTUs with any reference sequence is most often low compared to the sequence similarity of more abundant OTUs with the same taxonomic assignment. The fourth observation is that such seemingly redundant and less abundant OTUs almost consistently co-occur (i.e., are present in the same samples) with more abundant OTUs with a better taxonomic assignment. Based on these observations, it can be assumed that the majority of these low-abundant OTUs are in fact methodological and/or analytical errors, or rare (intragenomic) variants, which will cause inflated diversity metrics. Following from this assumption, the LULU algorithm is constructed to iteratively work though the OTU table to flag potential erroneous OTUs by employing the observed patterns of co-occurrence guided by pairwise similarity of centroid sequences of the OTUs. Thus, the algorithm takes advantage of the observed reproducible nature of extra/spurious OTUs and their sequence similarity to more abundant OTUs in the same samples and uses these features to infer their nature as errors (or true—but taxonomically redundant) variants of biological entities already represented in the table. After identification of these extra OTUs, they can be merged with their parent OTUs in order to preserve the total read count and reduce the OTU number of the table to a biologically reasonable level. The resulting table may be subjected to direct species richness metrics and other biodiversity analyses dependent on species-level OTU delimitation.

**The algorithm.** The LULU algorithm is a function written for R, accessible along with documentation on GitHub (https://github.com/tobiasgf/lulu). The workflow (Fig. 2) requires (1) an OTU table—a table in the form of a simple tab separated file with unique OTU-identifiers and their abundance across the investigated samples (samples as columns and OTU id's as rows)—and (2) a so-called match list—a list with the most similar OTUs (of the data set) for each OTU in the data set matches of OTUs. The match list is produced prior to the LULU curation by an external algrothm (e.g., BLASTn or VSEARCH). The match list should be the result of an internal matching of OTU sequences against each other, listing the best matches for each OTU. The match list should contain three columns: (i) the OTU-identifier of the focal OTU—the one being investigated as a potential error, (ii) the OTU-identifier of the potential parent, and (iii) a percentage measure of the similarity between the centroid sequences of the two OTUs. This measure of similarity may in principle be the result of any sequence comparison tool, in this study we have used BLASTn. The list may, and will most often, contain several rows for each OTU in the data set.

When passing the OTU table and the match list to the function, there are few user-selected parameters to consider. A minimum threshold (minimum_match) of sequence similarity for considering any OTU as an error of another can be set (default 84%). This setting should of course be adjusted so higher threshold is employed for genetic markers with little variation and/or few expected PCR and sequencing errors. Furthermore the user can specify the minimum co-occurrence rate (minimum_relative_occurrence)—i.e., the minimum acceptable fraction of presence of the potential error that can be explained by co-occurrence with the potential parent (default 95%). Lastly, it is possible to select (minimum_ratio_type and minimum_ratio) whether a potential error must have lower abundance than the parent in all samples (default), or if an error just needs to have lower abundance on average, and the ratio between the two. Choosing lower abundance on average over globally lower abundance will greatly increase the number of designated errors. This option was introduced to make it possible to account for non-sufficiently clustered intraspecific variation. However, it is generally not recommended to use this approach, as it will also increase the potential of clustering well-separated, but co-occurring, sequence similar species.

When passing the OTU table and the corresponding match list to the LULU function, OTUs of the OTU table are sorted first by decreasing occurrence (i.e. the number of samples containing that OTU) and subsequently by decreasing total read count. Thus, the OTU table can be curated from top down, so parents will be

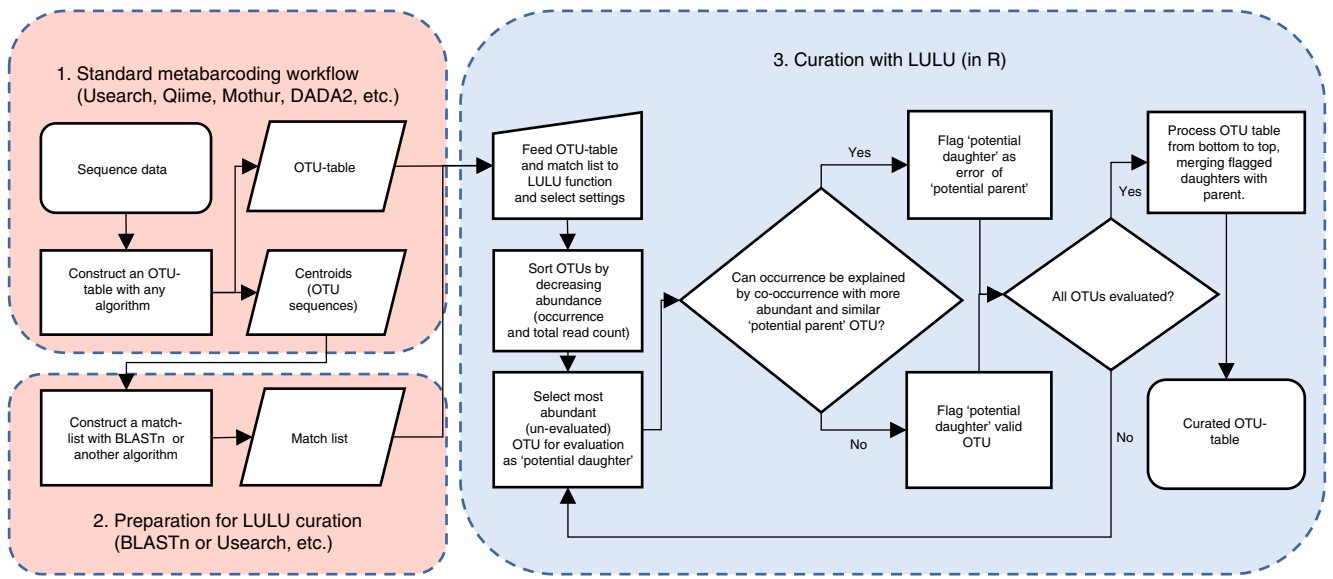

**Fig. 2** LULU curation workflow. (1) The user constructs an OTU table. (2) The user constructs a match list. (3) OTU table and match list is fed to the LULU algorithm

encountered before their respective derived errors. To evaluate the potential error state of an OTU, it is compared to all sequence similar OTUs—those appearing as hits in the match list—occurring in an equal or higher number of samples as the focal OTU. All potential parent OTUs satisfying these two conditions are selected for an evaluation of co-occurrence patterns. If the occurrence of the focal OTU can be explained by the simultaneous co-occurrence of a more abundant potential parent, the focal OTU is flagged as an error of that parent, and the algorithm moves to the next OTU on the table. If, however, the occurrence of the focal OTU cannot be explained by co-occurrence with a more abundant potential parent, the focal OTU is retained as a valid OTU.

After parsing the full OTU table, a new OTU table is constructed by merging read counts of errors with their designated parent OTU from bottom to top.

The function returns a list containing the curated OTU table along with the number and IDs of curated (retained) OTUs, the number of merged (daughter) OTUs and their IDs, information on which daughters were mapped to which parents, and information on user settings and runtime.

A very similar strategy is used for clustering of 'raw' sequences in the distribution based clustering algorithm (dbotu3) aiming at identifying ecologically distinct populations of bacteria and at the same time discarding ecologically redundant OTUs and errors. Dbotu3 differs from LULU by using un-clustered data —i.e. the distribution of reads among samples (≈ 0% clustering OTU table), another sequence dissimilarity metric (Levenshtein edit distance) in the form of a distance matrix based pairwise alignments of all sequences, another approach (the asymptotic likelihood ratio test) for evaluating whether two OTUs have similar distributions, and it is implemented as a python script.

**Plant survey data.** In order to validate the LULU algorithm, we collected data on vascular plants in two different ways: a reference data set using classical botanical identification and a metabarcoding data set based of soil samples. Both were collected in a set of 130 quadrats (site henceforth) dispersed across Denmark. The observational unit was a 40 × 40 m site. Sites were selected in an attempt to regularly cover the most important environmental gradients at the landscape scale, including naturalness of the habitat. 30 sites were allocated to cultivated habitats and 100 sites to natural and semi-natural habitats. The cultivated subset was stratified after major land-use types, while the natural subset was stratified according to gradients in soil fertility, soil moisture and ecosystem maturity. Saline and aquatic habitats were excluded, but mires and temporary wetlands were included.

The final set of 24 environmental strata consisted of the following six cultivated habitat types: three types of fields (rotational, grass leys, set aside) and three types of plantations (beech, oak, spruce). The remaining 18 strata were natural habitats, constituting all factorial combinations of: fertile and infertile; dry, moist and wet; open, tall herb/scrub and forest. These 24 strata were replicated in each of five geographical regions. We included a subset of 10 perceived biodiversity hotspots, selected by a poll among active natural history amateurs and professionals, but restricted so that each of the five regions were allocated two hotspots. The resulting number of sites was 130, evenly distributed across the five regions.

Each site was divided into four 20 × 20 m quadrants and the center of each of these—a 5 m radius circle (called a plot)—was in 2014 investigated thoroughly for vascular plants, and the results were compiled to a site species list for each site. In

2015–2016 a supplemental survey was conducted to produce more complete plant species list for the 130 sites. As the 2014 data is based on a standardized sampling strategy, these were used for richness correlations, whereas the 2014–2016 data were used for evaluation of site-wise taxonomic correspondence between OTUs and plant species.

**Sequence data.** The sequence data was generated by sequencing amplicons from DNA amplification of the nuclear ribosomal internal transcribed spacer region 2 (ITS2) with the primers S2F[30], and ITS4[31]. The ITS2 region was selected as it has been shown to be a good marker to separate and identify species[30,32,33]. DNA was extracted prior amplification from a subsample of soil collected from the set of sites (130 in total) described above. For each site, 81 equally spaced soil cores were collected, pooled and homogenized with a drilling machine (HILTI Cordless Combihammer), mounted with a clean mixing paddle. One bulk sample from the starting material of the 81 cores was taken and stored cold for further processing within 24 h.

From each homogenized sample, a subsample of 4 g was subjected to DNA extraction using PowerMax Soil DNA Isolation kit (MOBIO, Carlsbad, CA, USA), following the suggested protocol, after addition of 4 ml of 1 M suspension of CaCO3. An additional clean up step was performed on an aliquot, 100 µl of DNA extract, with the PowerClean DNA Clean Up Kit (MOBIO, Carlsbad, CA, USA). DNA concentrations were measured with Qubit dsDNA HS (High Sensitivity) Assay Kit (Invitrogen) and samples were normalized to a concentration of 1 ng/µl prior PCR amplification. PCR reactions contained 1 U/µl AmpliTaq Gold (Life Technologies), 0.625 µM of each primer, 0.83 mg/ml bovine serum albumin (BSA), 1X Gold Buffer, 2.5 mM of $MgCl_2$, 0.08 mM each of dNTPs and 1 µl DNA extract in a 25 µl total reaction volume. Thermocycling conditions used were an initial denaturation step of 5 minutes at 95 °C, followed by 32 cycles of denaturation of 30 s at 95 °C, 30 s at 55 °C, 60 s at 72 °C, and a final elongation at 72 °C for 7 min. Both forward and reverse primers were designed with 80 unique tags (MID/ barcodes) of 6–8 bp at the 5′-end using a restrictive dual-indexing approach. To obviate the error source of tag jumping resulting in mis-assignment of reads during demultiplexing, no primer tag (forward or reverse) was used more than once in any sequencing library and no combination of forward and reverse primer was reused in the study. Each sample was amplified three times using a different primer tag combination. PCR products were pooled for a total of 6 pools, each pool containing half of the samples from one PCR replicate and including one extraction blank and one PCR negative. PCR pools were purified with MinElute PCR purification kit (QIAGEN GmbH) and the length of PCR amplicons was verified on Bioanalyzer High-Sensitivity Chip (Agilent Technologies, Inc., Santa Clara, California, USA). Each of the 6 pools was built into separate sequencing libraries. Libraries were built using the TruSeq DNA PCR-Free Library Preparation Kit (Illumina), replacing all the manufacturer suggested clean up step (sample purification beads) with MinElute purification (MinElute PCR purification kit, QIAGEN GmbH). A final library purification was carried out to remove adapter dimers with Agencourt AMPure XP beads (Beckman Coulter, Inc., CA, USA). Sequencing was carried out on MiSeq (Illumina Inc., San Diego, CA, USA), at the Danish National High-throughput DNA Sequencing Centre, using one full 250 bp paired-end run. The data set contained 11,957,772 paired reads.

**Initial bioinformatic processing of sequence data**. For clustering with VSEARCH, SWARM, CROP and dbotu3, sequence data were processed in the same way. Paired reads were assembled with VSEARCH[22] (fastq_mergepairs) with default settings allowing staggered reads. Reads were demultiplexed and processed using a custom script intended for dual-indexed primers based on a procedure described here (https://github.com/frederic-mahe/swarm/wiki/Fred's-metabarcoding-pipeline#merge-paired-reads (accessed on 1 May 2017).). Tag and primer sequences were trimmed with CUTADAPT[34]. Reads with an expected error rate above 0.002, containing 1 or more N's, or with a length below 10 were discarded. Reads were dereplicated using VSEARCH. Reads from all three replicates were pooled for each sample, leaving 130 fasta files with dereplicated reads corresponding to each site. After merging and assigning reads to samples, the data set contained 6,629,544 reads. All bioinformatics steps can be found in the Supplementary Material and on GitHub.

**General validation approach**. To validate the LULU algorithm we used our plant survey data as ground truth for our amplicon data. OTU tables were produced with well established algorithms representing different approaches for OTU clustering and definition: (i) VSEARCH[22], representing greedy clustering algorithms similar to the commercial USEARCH[35], (ii) DADA2[13], based on a quality-aware model of Illumina amplicon errors, (iii) SWARM[23], an unsupervised single-linkage-clustering method, and (iv) CROP[24], an unsupervised Bayesian clustering method, and (v) DADA2 with subsequent VSEARCH clustering. The initial bioinformatics processing ensured that only high quality reads were kept (i.e., reads with an expected error rate above 0.002 or containing 1 or more N's were discarded). To ensure that we had removed as many errors as possible with the given tools prior to validating our algorithm, the implemented chimera removal tools of VSEARCH and DADA2 were employed in these analyses. All non-plant OTUs were discarded from all data sets to make the OTU richness comparable to field-observed plant richness (see below). We recorded the number of reads after clustering, taxonomic focussing and removal of singletons for each approach (Supplementary Table 5). The validation was focussed on the 97% clustering level, which is generally applied for ITS data, but we tested several other clustering levels in the range 95–100% (Supplementary Table 6). All the resulting OTU tables were curated with the LULU algorithm, and all un-curated and curated tables were then evaluated against the plant survey data with seven measures of correspondence: (i) site-wise OTU richness as a predictor of survey species richness, (ii) total OTUs richness vs. total survey species richness, (iii) taxonomic redundancy (proportion of OTUs with a taxonomic annotation already represented in the table), (iv) β-diversity (average α-diversity divided by γ-diversity), (v) distribution of reference database matches (best GenBank match of each OTU), (vi) taxonomic composition, and (vii) community dissimilarity indices before/after curation and compared to dissimilarity indices based on plant data. Furthermore we tested the distribution based clustering algorithm as implemented in dbotu3 as a one-step clustering method, as well as an alternative post-clustering algorithm.

**VSEARCH clustering**. Reads from all samples were pooled and dereplicated globally, chimeras were removed using uchime_denovo, clustering was done at 98.5, 98, 97, 96, and 95% dissimilarity levels, and for each clustering level an OTU table was produced and a file with representative OTU sequences.

**SWARM clustering**. Reads from all samples were pooled and dereplicated globally, and chimeras were removed with VSEARCH, clustering was done with SWARM[23] using d-values of 3,5,7,10,13, and 15, corresponding more or less to 99, 98.5, 98, 97, 96, and 95% clustering (Supplementary Table 1), and for each clustering level an OTU table was produced and a file with representative OTU sequences.

**CROP clustering**. Reads from all samples were pooled and dereplicated globally, and subsequently re-replicated with VSEARCH. Settings of CROP[24] were optimized for the actual read length and number of reads. Clustering was carried out with the parameters $l = 0.5$ and $u = 1.0$ corresponding more or less to 98%, and $l = 1$ and $u = 1.5$ corresponding more or less to 97%, and and $l = 1.5$ and $u = 2.5$ corresponding more or less to 95% dissimilarity levels (Supplementary Table 1). Reads were mapped against the defined OTU representative sequences using VSEARCH at levels 98, 97 and 95% respectively and OTU tables were produced.

**DADA2 processing**. DADA2[13] requires sample-wise libraries—i.e., one pair of fastq files per sample where reads do not include primers or tags. As our laboratory methods rely on multiplexing of several samples in each library, we constructed a script for demultiplexing without merging. Also, DADA2 relies on separate processing of forward and reverse reads. Our multiplexing method relies on annealing of adapters to amplicon pools, which means that half of the reads will be inserted in reverse direction. DADA2 is based on the distribution of errors, and as the distribution of errors cannot be assumed to be identical between R1 and R2 reads, we chose to process the sense and anti-sense reads separately, and merge the results in the end.

Paired reads were demultiplexed and processed using a custom script based on CUTADAPT[34]. Forward and reverse reads were demultiplexed separately. Before processing with DADA2, matching of forward and reverse reads was ensured with DADA2. Processing of the reads then followed the procedure outlined here (http://benjjneb.github.io/dada2/tutorial.html, accessed 1 May 2017). The chimera removal tool of DADA2 (removeBimeraDenovo) was employed. Lastly the tables produced for sense reads and anti-sense reads were merged. OTU sequences were extracted with R. After DADA2 processing the data set contained 5,725,783 reads.

**DADA2 with subsequent VSEARCH clustering**. DADA2 has been shown to accurately identify highly resolved microbial communities and produce few spurious sequences[13]. As we assumed that the pure DADA2 approach will identify sub-specific and intragenomic types of ITS2 sequences[13], and thus result in an inflation of the richness, we devised an approach with subsequent clustering with VSEARCH. Reads were extracted with abundance information sample wise for 130 fasta files with dereplicated reads corresponding to each site. The extracted reads were then subjected to the VSEARCH approach outlined above, clustering at 98.5, 98, 97, and 95%.

**Taxonomic assignment and filtering**. To make sequence data maximally comparable to reference data, we assigned taxonomy and filtered out non-plant OTUs from each table. To optimize the process, we processed all OTUs together. Centroids from all 20 tables (6 SWARM, 5 VSEARCH, 3 CROP, 1 DADA2, 5 DADA2 + VSEARCH) were pooled and dereplicated. The best GenBank matches for each OTU were acquired using BLASTn[36] (with settings -qcov_hsp_perc 90 -perc_identity 80), keeping up to 20 matches pr. OTU. For each OTU, all hits, from the best match and down to matches half a percent (0.5%) lower than the best, were retained, and the most commonly assigned taxonomic id was identified, and the taxonomic path (kingdom, phylum, class, order, family, genus, species) was acquired from the NCBI taxonomy. The ingroup OTUs were identified as belonging to Streptophyta, but excluding Chlorophyta, Sphagnopsida, Jungermanniopsida, Bryopsida, and Polytrichopsida. With the ingroup OTUs defined, the 20 OTU tables and centroid files were filtered to contain only ingroup OTUs.

**LULU curation**. The OTU tables were then curated with LULU. For each of the centroid files corresponding to one of the OTU tables, match lists were produced by making a blast database of the OTU sequences (makeblastdb -in centroids.fasta -parse_seqids -dbtype nucl) and subsequently making a blast search against the database with the same reads (blastn -db centroids.fasta -num_threads 50 -outfmt '6 qseqid sseqid pident' -out matchlist.txt -qcov_hsp_perc 80 -perc_identity 84 -query centroids.fasta). Each set of OTU table and match list were then used as in input for the LULU algorithm, and the curated tables and OTUs were used for comparison with the un-curated/raw tables.

**Site-wise OTU richness as a predictor of plant richness**. As the most important measure of validity, we used a comparison of the OTU richness with the 'real' richness (the observed vascular plant richness). DNA extracted from soil may contain DNA from more plant species than can be observed in a classical survey. Soil may harbor DNA from plants that are not apparent or biologically active at the time of investigation as these may be represented by e.g. pollen, seeds, etc., and soil particles can bind and preserve DNA from species no longer present. Furthermore, morphologically defined species may harbor cryptic but genetically separate species, and also the investigated ITS2 region is known to have varying levels of intraspecific and intragenomic variation that is difficult to accommodate within an universal clustering level. However, the sampling of this study was carried out in a temperate environment with productive soils, and we assume that these DNA remains will be present in too low abundance and too fragmented to amplified with the selected primers (targeting a region of 300+ bp on average) to pose a problematic contribution to the sequence pool. Considering this and taking into account, that the soil sampling only covers approximately 0.01% of the soil surface of the $40 \times 40$ m sites, we expect the sequencing approach to underestimate the true diversity, because many species occurring with few or small individuals only are likely to be missed. For each method at each clustering level, the OTU richness of each of the 130 sites was compared with the observed vascular plant richness for pre- and post-curation tables. As Initial inspection of quantile-quantile plots of residuals indicated that normality was a fair judgment, Pearson correlations were assessed (Fig. 1a, Table 1, Supplementary Figs. 1 and 2).

**Total OTU richness vs. total survey species richness**. We expected that the sequencing approach would identify fewer plant OTUs than the total number of plant species recorded in the reference data, as the soil sampling covered only a proportion of the soil surface. In the survey we observed a total of 564 plant species (approximately one third of the naturally occurring plant species in Denmark). Thus, we compared the total number of OTUs identified by each method to this number (Fig. 1b, Supplementary Figs. 3 and 4).

**Taxonomic redundancy**. We evaluated taxonomic redundancy of the raw tables produced with each method and the effect of curation on this measure. This was done by calculating the proportion of OTUs with a redundant taxonomic assignment—i.e. the number of OTUs assigned a species name already present in the

table divided by the total number of OTUs (Fig. 1c, Supplementary Fig. 5). We tested the availability of *ITS2* sequence data for plant species registered in Gen-Bank. This was done by searching for the combination of taxon name (at species level, i.e., removing sub-specific taxonomic levels) using the search term "taxon_name[Organism] AND internal_transcribed_spacer_2[misc_feature]". 501 of the 564 species names registered in the survey had at least one hit, corresponding to a coverage of 88.8%, Thus, several OTUs assigned to the same taxonomic identity is likely to be an indication of erroneous OTUs or in other ways taxonomically redundant OTUs for species level investigations. We did not expect a redundancy of 0% for several reasons. Many plant species are still not represented by sequences in GenBank and, if present as OTUs at a site, they will be assigned a name of a close relative, which may be already present in the data. Furthermore, several sequences are incorrectly annotated in GenBank, so a perfect match may still carry a wrong (and possibly redundant) annotation. Also, if intragenomic variation is not absorbed by the chosen clustering level, and the dominant *ITS2* type varies between populations, the error pattern used by LULU will not be satisfied resulting in redundancy.

**β-diversity**. From a realistic OTU definition, we expected a β-diversity of OTUs approaching that of the observed β-diversity of vascular plant species. To evaluate this, we used the simple β-diversity measure of total richness divided by the average richness pr. site (average α-diversity divided by γ-diversity). This was done for all curated and un-curated tables and compared to the same measure for plant species, which was 17.23 (Fig. 1d, Supplementary Fig. 6).

**Distribution of reference database matches**. To further substantiate that LULU identifies true errors, we estimated the likely error state of each OTU by looking at the best match (%) against GenBank. Although the LULU algorithm is independent of a reference database, we can take advantage of the fact that GenBank is relatively well populated with sequence data assigned to plant species observed in our survey (see above), to evaluate the curation algorithm. Thus, we assume that OTUs representing true biological species are more likely to have a perfect or near-perfect match on GenBank, whereas PCR or sequencing errors (and very rare biological variants lying outside the clustering limits) are more likely to have a non-perfect match. We compared the distribution of best reference database matches for retained vs. discarded OTUs for each method and level (Fig. 1e, Supplementary Fig. 7). We did not expect all 100% matches to be retained, as some may be taxonomically redundant (intraspecific/intragenomic variants already represented by another OTU), and likewise we did not expect all non-perfect matches to be discarded, as they may well represent species or intraspecific variants not present in GenBank. But, for an effective curation, we expected the majority of the matches for the retained OTUs to be around 100%, and the density of the discarded to be lower.

**Taxonomic composition**. To further evaluate whether LULU retained the 'correct' OTUs, we compared the taxonomic composition of OTUs to the plant survey data for each site (Table 2). For each site we calculated: (i) imperfect matches: the proportion of OTUs with an imperfect (less than 100%) reference database match, (ii) recaptured species: the proportion of OTUs with a 100% reference database match and a unique taxonomic annotation that corresponded to a plant species recorded for that site in the survey, (iii) unregistered species: the proportion of OTUs with a 100% reference database match and a unique taxonomic annotation that corresponded to a plant species not recorded for that site in the survey, (iv) redundant species: the proportion of OTUs with a 100% reference database match but a redundant taxonomic annotation (Table 2). Furthermore we calculated the proportion of recaptured OTUs from the initial OTU algorithm that were lost during curation. We postulate that a valid curation should primarily result in a smaller proportion of imperfectly matching OTUs and an increased proportion of recaptured species, without losing a large proportion of recaptured OTUs. The reference database (GenBank) contained ITS2 data on 88% of the species recorded in the survey. Thus, if the OTUs of a site constitute a perfect subsample of those recorded in the survey, we would expect the proportion of OTUs classified as 'imperfect matches' to be around 12%, and the proportion of 're-captured' to approach 88% with a perfect OTU delimitation and curation. However, ITS2 shows intraspecific variation, and not all the species in the 88% GenBank coverage will be a perfect match, so the proportion of imperfect matches must be assumed to be somewhat higher. We expect some plants to be missed by the survey, although they were actually present (at least as DNA), but detected by the molecular methods, resulting in the proportion of OTUs classified as 'recaptured' to be lower.

**Community dissimilarity**. To further test the validity of the lulu curation, we investigated the effect on community dissimilarity estimates. Assessment of composition and turnover is driven by dominant and abundant species, and is relatively insensitive to errors, rare species, and low-abundance species, and thus, we expected both curated and un-curated OTU-tables to be adequate for estimating plant community dissimilarity. Thus, we hypothesized that (1) a valid curation would have no major impact on dissimilarity measures based on un-curated vs curated OTU tables, and (2) that a valid curation could not make the correlation between dissimilarity measures based of survey data and OTU data larger by

curation. To test these hypotheses this we estimated community dissimilarity of all 40 OTU tables and the plant survey data with the Bray-Curtis metric using the vegdist function as implemented in the r package vegan[37]. Dissimilarity matrices were calculated for binary (presence/absence) data for all tables and for the OTU tables also with hellinger transformed abundance (read count) data to see whether read abundance would yield better metrics. To test hypothesis 1, community dissimilarity matrices based on the 20 uncurated OTU tables were compared to dissimilarity matrices based on their curated counterparts using the mantel test with Pearson correlation using 999 permutations. This was done for both for binary and abundance data versions (Supplementary Table 1). To test hypothesis 2, dissimilarity matrices for all 40 OTU tables (20 uncurated, and 20 curated) were compared individually to the dissimilarity matrix for plant survey data using the mantel with Pearson correlation and 999 permutation, postulating the a valid curation cannot result in a lowered Mantel *r*-statistic (Supplementary Table 2).

**Singleton removal compared to post-clustering curation**. A traditional approach for reducing the number of PCR and sequencing errors are to remove singletons, despite singletons may represent real species. As a lot of singletons can be assumed to be errors, we wanted to compare effect of singleton removal of our data and compare to post-clustering curation with LULU. We removed singletons (observations with read counts of one) from all initial tables produced with VSEARCH, SWARM, DADA2 and CROP, and subjected the resulting tables to the same metrics as the un-curated and LULU curated tables and compared the results, to test whether this simple error-removal strategy could perform similar improvements of biodiversity metrics. (Supplementary Figs. 8–15, Supplementary Tables 3 and 4).

**Dbotu3 as alternative to LULU for post-clustering curation**. We tested the performance of dbotu3 as an alternative post-clustering algorithm to LULU. Although intended to work as a 'one-step' clustering algorithm with the aim of identifying ecologically distinct populations and at the same time discarding ecologically redundant OTUs and errors, the data processing strategy of dbotu3 is related to the post-clustering curation of LULU. Thus, we wanted to test this algorithms performance as an alternative to LULU for post-clustering curation. To do this we applied dbotu3 to the same set of 20 initial OTU tables and corresponding centroids as used in the validation of LULU (see above). Dbotu3 was applied with a genetic dissimilarity maximum of 16%, larger than the suggested 10% in the online manual (http://dbotu3.readthedocs.io/en/latest/, accessed June 17, 2017), but corresponding more or less to the 84% dissimilarity cutoff (minimum_match) employed in the LULU curation of the other data sets. We used two different approached as suggested in the online manual First, we analyzed the data using an abundance criterion of 0 (python dbotu3.py-dist 0.16-abund 10)—an approach aiming at accounting for only sequencing error. Second, we used an abundance criterion of 10 (python dbotu3.py-dist 0.16-abund 0), aiming at merging ecological populations. Results were benchmarked against the results of LULU for several metrics (Supplementary Figs 8–15, Supplementary Tables 3 and 4).

**Distribution based OTU clustering**. We also tested the performance of the distribution based clustering algorithm implemented in dbotu3 as a 'one-step' clustering method and compared the results with our plant survey data to see whether it could serve as an 'all-in-one' tool for clustering and curation compared to initial clustering and post-clustering curation. Reads from all samples were pooled and dereplicated globally, and reads from each sample was mapped against these representative reads (centroids) to produce an 0% clustering OTU table, which is the input for dbotu3 along with the centroid sequences. The 0% clustering table contained 722,493 OTUs. Dbotu3 was applied to the table and corresponding centroids using the same settings (genetic cutoff 16%, and $a = 0$ and $a = 10$) as above. OTUs were subsequently restricted to plant OTUs, and the results were compared to those from of the approaches employing initial clustering and subsequent curation. (Supplementary Fig. 16, Supplementary Tables 2–4).

**Curation effect on selected plant genera**. To evaluate the more detailed effects of curation, we selected a number of plant genera for a closer look at the effect of curation. We selected genera with (1) high levels of occurrence and abundance in the data, (2) relatively stable taxonomy and name use, and 3) for which reference data were good (occurrence in the sites and the region in general). For each genus, we plotted the abundance (read count) and best match of all OTUs assigned to the genus for all combinations of clustering method and clustering level. We then evaluated the curation effect against ground truth data, i.e. occurrence data from the plant survey (Supplementary Figs 17–32). We selected the following genera for evaluation: *Acer*, *Alnus*, *Avenella*, *Calamagrostis*, *Calluna*, *Centaurea*, *Cerastium*, *Erica*, *Fagus*, *Filipendula*, *Holcus*, *Littorella*, *Lysimachia*, *Menyanthes*, *Plantago*, *Poa*, and *Potentilla*.

**Data availability**. The LULU R package is open source and available on GitHub (https://github.com/tobiasgf/lulu), where instructions for installation and use also can be found, along with scripts and R Markdown files used for the data analyses. The sequence data is available from the Dryad Digital Repository: http://dx.doi.org/10.5061/dryad.n9077.

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

## Acknowledgements

This work was supported by a grant from VILLUM foundation (Biowide, VKR-023343). Irina Goldberg is thanked for carrying out the plant survey, Ida Broman Nielsen and Anne Aagaard Lauridsen are thanked for contributing to the lab work.

## Author contributions

T.G.F., R.K., H.H.B., R.E., A.K.B., A.J.H. designed the project and sampling design; C.P., T.G.F. performed the laboratory work; T.G.F. designed and implemented the algorithm; T.G.F. performed the analyses; T.G.F., R.K., H.H.B., R.E., A.K.B., C.P., A.J.H. wrote the paper.

## Additional information

**Competing interests:** The authors declare no competing financial interests.

