## [Peer Review File · Nature Communications]

Reviewers' comments:

Reviewer #1 (Remarks to the Author):

Review of "Reliable biodiversity metrics from post-clustering curation of amplicon data"

In this manuscript Froslev and colleagues introduce a new method (LULU) for identifying artefactual or redundant operational taxonomic units (OTUs) from high-throughput metabarcoding data. LULU uses patterns of co-occurrence combined with sequence similarity to flag OTUs as either artefactual or redundant with other OTUs, and condenses the OTU table by merging those OTUs together. This method is validated on an ITS metabarcoding dataset collected from soil, where the taxonomic composition of each sample was also evaluated by a botanical survey of the local vascular plants. LULU is shown to improve the correspondence between HTS measurements of community composition and the results of the botanical survey across several types of metrics. Furthermore, several bioinformatic methods for producing the OTU table from raw sequencing data are evaluated, and a recommendation is made on the best approach.

This manuscript is well-written, the analysis has multiple strengths, but there is one critical flaw that must be addressed before it can be published. The strengths include (1) robust benchmarking of metabarcoding methods in the context of surveying plant biodiversity, (2) evaluation of state-of-the-art methods that are seeing rapid uptake, especially VSEARCH and DADA2, and (3) a very strong validation dataset that spans a wide range of ecological conditions and has a strong ground-truth component from the botanical survey.

The critical flaw is that the LULU method is less novel than realized by the authors, there is a prior method in the literature that also uses patterns of co-occurrence and sequence similarity to merge OTUs in a similar fashion: Preheim SP, Perrotta AR, Martin-Platero AM, Gupta A, Alm EJ. Distribution-based clustering: using ecology to refine the operational taxonomic unit. *Applied and environmental microbiology*. 2013 Nov 1;79(21):6593-603.

In order to be published, the paper must be revised to acknowledge and discuss the distribution-based clustering method. This reduces the novelty of the introduced method. However, provided a suitable revision is made, I believe the strengths of this paper make it of significant interest to the research community and warrant publication in *Nature Communications*. Plant and ITS metabarcoding analyses are underserved when it comes to benchmarking and validating methods, so the results and recommendations here are quite useful. The dataset compiled and analyzed here is excellent. And co-occurrence methods like LULU are timely given the increased uptake of exact sequence methods like DADA2 where intragenomic variation is more likely to be captured (which is something that could be highlighted even a bit more in the manuscript).

Itemized comments:

L88-89: VSEARCH does not implement a method similar to UPARSE, rather it is similar to the earlier UCLUST method.

L108-110: But too far below unity is worse, no? While I get the point that a slope $\gg 1$ almost certainly indicates lots of artefacts, it's not obvious to me that the lower the slope, the better, which is what this sounds like.

L159: "simple Beta-diversity measure": Introduce the measure immediately in the main text, at least in brief. This is important because "beta-diversity" is often used in a different way, as a synonym for community dissimilarity, in the bacterial metabarcoding literature.

L166: "(except...)": This is confusing, as the "except" reads as it applies to the whole sentence, rather than just the "ranging from 19.1 to 26.5" clause.

L185-187: Again, its best if you can briefly describe the simple dissimilarity measure you are using right here (eg. "number of unshared genera" or something like that).

L201-208: Good discussion of the potential pitfalls of using HTS data exactly as the data from traditional ecological studies.

L233-L240: I can't quite follow what is being said here. What is "alternating proportions"? If I had to guess, some plants have intragenomic ITS variants in equal proportion, and it stochastically alternates as to which has more reads? This should be clarified, in particular because I think that the potential use of LULU-like algorithms on exact sequence data (eg. DADA2) has the potential for broad use given intragenomic variation in marker-genes exist across all domains.

L289-293: confusing writing

L294-L296: Also mention intra-genomic variation?

"The algorithm": Is it possible to make the method and all necessary tools installable via an R package manager? It is not necessary for publication, but it has the potential to increase the impact of the work in the long-term.

L329-332: Unclear writing. Occurrence is the fraction of samples in which it is present?

L342-343: What is "occurrence frequency"?

"Plant Survey Data": The broad range of environments included here is impressive.

L456-458: How were non-plant OTUs discarded?

L506: When citing a web page, it is recommended to date it (especially for version-controlled web sites like this one, as then the exact same page can be retrieved later on).

L514-515: The DADA2 method does correct PCR errors.

Table 1: If space permits, I would like to see the slope/intercept values included in this table as well.

Supp. Figure 1a/b: Need a high-resolution version of these figures. Can't read any of the fit parameters, which I'm quite interested to see.

Reviewer #2 (Remarks to the Author):

In this paper they have developed a post-clustering curation method of pre-processed metagenome sequencing data by merging erroneous OTUs into a major cluster. They have validated this method using plant metagenome data and improved the accuracy of the richness estimates. However,

algorithmic details are unclear and support of their hypothesis is validated in only special case which is not generalizable. Substantial revision is needed to be acceptable in the journal.

General comments

1. The authors stated that the algorithm does not require arbitrary cut-off levels. However, some user-selected default parameters exist in the algorithm. The author must clearly explain this statement and discuss the advantage over previous methods of applying arbitrary level to discard erroneous OTUs.
2. It is unclear to me that this present method is the only curation method available for amplicon sequencing data. There exists a paper (doi:10.1128/AEM.01043-13) using dual indexing strategy. The authors should discuss what has been done before and compare the strategies.
3. In comparing total OTU richness versus total survey species richness, the CROP algorithm was the only method to underestimate the total richness and did not improve much after curation. It would be interesting to discuss what drives this result.
4. The authors insist that LULU algorithm greatly improve the estimates of the accuracy of richness derived from amplicon sequencing data in various measures such as alpha or beta diversity. Their algorithm was developed based on well-studied organism. It would be interesting to know if this post-clustering algorithm applies well to the other organism groups. Further evaluation at other application would be helpful to validate the results.
5. In algorithm section, description is too vague to understand. Although they said any sequence comparison tool can be used for measuring similarities, the tool they have used in this study needs to be specified. In addition, the process of making match list is not properly described how focal OTUs can be found in this process and how to find potential parent OTUs. Even supplement note does not have information for generating this list. Furthermore, this section seems to have mixed information of description of how to use this LULU and LULU's analysis step. It will be better if authors separate these two in different section or upload manual on github page.

Minor comments

1. The term "OTU" needs to be defined at the first time.
2. All the supplementary figures lack resolution that was hardly recognizable.
3. There is only one figure summarizing the result. The paper focuses on the algorithm and needs the schematic figure flow of their core logic. In the methods section, the algorithm is written too general and it was hard to know the distinctive advantage over other methods. It needs to be clarified and be more specific to the readers.
4. For the most, LULU's github page mentioned in this manuscript is not available currently and needs to be fixed.
5. There are some minor typos throughout the paper.

Reviewer #3 (Remarks to the Author):

Frøslev et al. present an algorithm called 'LULU' that attempts to reduce impacts of DNA sequencing errors on estimates of species richness in DNA metabarcoding surveys. Relying on assumptions about the signatures of DNA sequencing errors in these datasets, the algorithm is shown to reconcile estimates of species richness obtained from field survey data of vascular plants with the richness of plant DNA in soil cores at the same sites. The authors interpret this as evidence that errant DNA

sequences have been culled from the dataset.

The problem of accurately estimating species richness from sequence-based surveys is widespread and needs to be tackled. I commend the authors for addressing this issue in a comparative approach, but I have several questions/concerns that still need to be addressed so that the reader can gauge the authors' assumptions/interpretations in comparison to the alternatives.

General comments:

1. In "validating" the algorithm, the authors suggest "true" sequences are retained and "false" sequences are discarded. Results are consistent with this suggestion, but this cannot definitively be determined from correlations with plant surveys; experiments using "mock communities" of known composition would be required.
2. The second half of the introduction focuses heavily on methods but lacks detail, leaving the reader to interpolate how the main results of the manuscript relates to the broader problem being addressed. See several comments below.
3. The authors are critical of several previously used strategies to "curate" similar datasets—such as arbitrary decisions to cull singletons from OTU tables—and rightfully so. However, they did report how the inferred level of species richness based on these decisions differ from the output of LULU; instead, they reported how a set of uncurated OTU tables compare to the output of LULU. There are two potential issues:
 - (a) Without experimentally classifying OTUs as "true" or "false" (e.g., from mock communities), any method that reduces a bad overestimate of richness to a more reasonable value—including arbitrary culling of singletons or inaccurate culling of some "true" sequences—can appear to improve correspondence with ecological surveys. Thus, the reported correlations may be suggestive of increased accuracy after applying LULU, but causation remains unclear.
 - (b) Most researchers go beyond clustering OTUs in their attempts to remove errant sequences from DNA metabarcoding data. Therefore, it would be better to compare the output of LULU to both the raw OTU tables and to several commonly employed sets of assumptions (e.g., culling singletons) and/or existing algorithms designed to remove errors and denoise these types of datasets. By comparing LULU's output to OTU tables without any additional effort to remove errors, the authors may inadvertently present a false dichotomy.

Title: Is "post-clustering" a very searchable/informative term to use?

Line 15: I suggest using "DNA metabarcoding" to match the lexicon.

Line 16: Is "LULU" an acronym that should be defined?

Line 19: Be careful to provide context for the word "barcode". There are differences in implied meaning between research communities, as some refer to the standard markers used by the "barcode of life" (e.g., COI) while others refer to molecular identification tags (i.e., "MID-tags").

Line 20: Is a minimum number of samples needed to make strong inferences based on co-occurrence patterns? The abstract may not be the appropriate place to address this question, but it needs to be addressed clearly.

Line 20-21: Specifically, what types of markers, organisms, or environmental gradients were sampled?

Lines 25-27: Although LULU may not require a reference database to run, please be careful to avoid

implying that reference databases are not needed/useful for assessing the accuracy or completeness of DNA metabarcoding studies. Indeed, your own assessments employ one.

Line 35: "poorly studied" – would it be more constructive to describe the specific ways in which previous research has not completely addressed this issue?

Line 53: "many spurious OTUs" – could the results of this reference be presented more precisely, and perhaps in comparison to the "38%" estimated in the previous reference?

Lines 56-58: Although it is true that researchers do this, it is also true that these decisions should depend on the marker and taxonomic group they employ.

Lines 58-65: Good – this concisely captures a major problem: the ecological reality of rare species needs to be accounted for in algorithmic approaches to curating datasets. Please also note that the presence of DNA in a substance (e.g., soil) does not imply that it corresponds to an organism with any biological function in that substance. Thus, there may be "true" DNA sequences in a sample that should not be targeted for removal by algorithms but that also may lack any relationship with other ecological data (e.g., local plant surveys).

Lines 77-84: Much more detail on the surveys are needed: where in the world were these surveys; how were they sampled; what are the expected levels or ranges of richness based on these surveys; across which taxa are these ITS markers "comprehensive" in coverage; what is the nature of the "reference data" for vascular plants (i.e., presence vs. absence or abundance)? Although lengthy details can be saved for the methods, the reader will struggle to understand the results as written.

Lines 82-84: By what criteria were GenBank data deemed "adequately populated"? Were all of the species recovered in the plant surveys also present in the database? If not, how few?

Line 86: Do the terms "curated" and "un-curated" refer to datasets that have been subjected to the LULU algorithm or not, respectively? I suggest stating this clearly, and being consistent throughout the text.

Lines 86-97: This paragraph is methods-heavy—indeed sections of it seem to be lifted from text that also occurs within the methods section. I suggest making this the first paragraph of the "Results" section, and using it to define the structure of the results section, in which multiple comparisons are presented sequentially under subheadings (even though all of these subsections refer to Figure 1).

Lines 95-96: Please give a concise definition of "Taxonomic redundancy" and "distribution of reference database matches" to save the reader from having to flip through the methods section.

Line 108: assumption #3 – this assumption needs to be supported rigorously, or perhaps reconsidered, because opposite expectation could also be supported. For example, (a) soil particles can bind and preserve plant DNA long after local plant composition changes (~years to centuries), (b) soil can contain DNA of plants that are not locally apparent (e.g., from pollen, dormancy, etc.), (c) unique ITS sequences within the soil may (truly) be more numerous than the plant species that contributed to them, due to the concerted evolution and intragenomic variation that can plague ITS-based studies. While the presented expectation may hold under some circumstances, this is likely to be scale-dependent and the reader does not know the scale of your sampling regimes at this stage.

Line 131: Good – the application of LULU may not only "improve" the selected measures, but in 3 / 4 of comparisons the resulting counts fell below your 'maximum' criterion of 564 species. This could be

stated more directly.

Line 141: I suggest trying to justify this approach with citations and/or by presenting a separate analysis of the available GenBank data.

Line 159: "simple beta-diversity measure" – which one? Was it a weighted or unweighted metric?

Line 169: "Assuming that Genbank is well populated" – this should be easy to check by directly comparing vegetation survey data to GenBank data.

Lines 313-314: In what format? Does this format match an output that can be obtained directly from each of the OTU-clustering programs utilized?

Line 339: Seems like there is a word missing or out of place in this sentence.

Line 393: What is known about the taxonomic "coverage" and species-level "specificity" of these markers? (See, e.g., Ficetola et al. 2010 for discussion of coverage and specificity)

Line 410: subscript the 2 in "MgCl₂"

Lines 439-440 (and elsewhere; 505-506): If one goal of the manuscript is to provide a method for wide use by the research community, I suggest minimizing "custom" scripts in your methods or at least providing relevant references to published/commercial strategies for accomplishing the same task.

Lines 454-456: Analyzing data available in GenBank could also test this assumption. For plant species known from the plant surveys, is this assumption upheld consistently? If too few of the local species have representative ITS sequences in GenBank, this should be acknowledged, but the specificity of the marker could still be evaluated (e.g., using "ecoPCR" as in Ficetola et al., 2010).

Line 530: "0.51%" – is this value meant to be a percent or a proportion?

Lines 533-536: What proportion of sequence reads and unique sequences were retained within the array of samples?

Lines 580-582: Also note the potential for intragenomic variation using ITS.

Lines 614-616: "We postulate...removal of OTUs cannot result in... increased dissimilarity" – why not?

Lines 618-620: "An evaluation of community composition relies heavily on correct annotation" – this is not necessarily true as written. It is certainly possible to compare the composition of two species x site matrices in the complete absence of Latin binomials. I would contend that the estimated dissimilarity of sites based on a standard metric (e.g., Bray-Curtis) would be highly concordant between comparisons based on curated and uncurated OTU tables. This conjecture could be evaluated using Mantel tests. I suggest the authors present such analyses because, at the very least, it brings into consideration whether the putative sequencing errors that LULU was developed to remove might impact estimates of species richness (the focus of the manuscript's introduction) as well as ecological dissimilarity (which is likely less sensitive to these sequencing errors).

Line 656: "trend" vs. "trends"

Reviewers' comments:

Reviewer #1 (Remarks to the Author):

Review of “Reliable biodiversity metrics from post-clustering curation of amplicon data”

In this manuscript Froslev and colleagues introduce a new method (LULU) for identifying artefactual or redundant operational taxonomic units (OTUs) from high-throughput metabarcoding data. LULU uses patterns of co-occurrence combined with sequence similarity to flag OTUs as either artefactual or redundant with other OTUs, and condenses the OTU table by merging those OTUs together. This method is validated on an ITS metabarcoding dataset collected from soil, where the taxonomic composition of each sample was also evaluated by a botanical survey of the local vascular plants. LULU is shown to improve the correspondence between HTS measurements of community composition and the results of the botanical survey across several types of metrics. Furthermore, several bioinformatic methods for producing the OTU table from raw sequencing data are evaluated, and a recommendation is made on the best approach.

This manuscript is well-written, the analysis has multiple strengths, but there is one critical flaw that must be addressed before it can be published. The strengths include (1) robust benchmarking of metabarcoding methods in the context of surveying plant biodiversity, (2) evaluation of state-of-the-art methods that are seeing rapid uptake, especially VSEARCH and DADA2, and (3) a very strong validation dataset that spans a wide range of ecological conditions and has a strong ground-truth component from the botanical survey.

The critical flaw is that the LULU method is less novel than realized by the authors, there is a prior method in the literature that also uses patterns of co-occurrence and sequence similarity to merge OTUs in a similar fashion: Preheim SP, Perrotta AR, Martin-Platero AM, Gupta A, Alm EJ. Distribution-based clustering: using ecology to refine the operational taxonomic unit. Applied and environmental microbiology. 2013 Nov 1;79(21):6593-603.

RESPONSE: Thanks. We are glad that our attention was drawn to this publication. We acknowledge that this method is similar in structure to our method, despite different objectives. We have now included a thorough comparison and discussion of this approach in our manuscript, and changed the terminology accordingly to acknowledge that this related approach has been introduced earlier. We have tested the distribution based clustering algorithm dbotu3 on our (raw) data and included that in the manuscript. The processing time was 10.5 - 17 days(!), and the results

were not very good in comparison with the curated tables from the other approaches. Although dbotu3 was not intended for that purpose, we have also tested whether the dbotu3 algorithm can perform a similar post-clustering curation as LULU and function as an alternative, and included these tests in the manuscript. (Although LULU outperformed dbotu3 in almost all comparisons, dbotu3 did a good curation job).

In order to be published, the paper must be revised to acknowledge and discuss the distribution-based clustering method. This reduces the novelty of the introduced method. However, provided a suitable revision is made, I believe the strengths of this paper make it of significant interest to the research community and warrant publication in Nature Communications. Plant and ITS metabarcoding analyses are underserved when it comes to benchmarking and validating methods, so the results and recommendations here are quite useful. The dataset compiled and analyzed here is excellent. And co-occurrence methods like LULU are timely given the increased uptake of exact sequence methods like DADA2 where intragenomic variation is more likely to be captured (which is something that could be highlighted even a bit more in the manuscript).

RESPONSE: Thanks for the positive remarks. See below for specific responses.

Itemized comments:

L88-89: VSEARCH does not implement a method similar to UPARSE, rather it is similar to the earlier UCLUST method.

RESPONSE: Of course. It has been corrected.

L108-110: But too far below unity is worse, no? While I get the point that a slope $\gg 1$ almost certainly indicates lots of artefacts, its not obvious to me that the lower the slope, the better, which is what this sounds like.

RESPONSE: This has been clarified now.

L159: “simple Beta-diversity measure”: Introduce the measure immediately in the main text, at least in brief. This is important because “beta-diversity” is often used in a different way, as a synonym for community dissimilarity, in the bacterial metabarcoding literature.

RESPONSE: Good idea. Has been added now.

L166: “(except...)”: This is confusing, as the “except” reads as it applies to the whole sentence, rather than just the “ranging from 19.1 to 26.5” clause.

RESPONSE: This has been clarified now.

L185-187: Again, its best if you can briefly describe the simple dissimilarity measure you are using right here (eg. “number of unshared genera” or something like that).

RESPONSE: A brief description has been added.

L201-208: Good discussion of the potential pitfalls of using HTS data exactly as the data from traditional ecological studies.

RESPONSE: thanks...

L233-L240: I can't quite follow what is being said here. What is “alternating proportions”? If I had to guess, some plants have intragenomic ITS variants in equal proportion, and it stochastically alternates as to which has more reads? This should be clarified, in particular because I think that the potential use of LULU-like algorithms on exact sequence data (eg. DADA2) has the potential for broad use given intragenomic variation in marker-genes exist across all domains.

RESPONSE: This has been clarified now, and extended with a suggestion to implement LULU-like algorithms in pipelines like DADA2

L289-293: confusing writing

RESPONSE: Agree. It has been reformulated.

L294-L296: Also mention intra-genomic variation?

RESPONSE: Yes, done.

“The algorithm”: Is it possible to make the method and all necessary tools installable via an R package manager? It is not necessary for publication, but it has the potential to increase the impact of the work in the long-term.

RESPONSE: It is available as an installable package on GitHub. The information necessary to install it is now added to the manuscript (and not only in the markdown files).

L329-332: Unclear writing. Occurrence is the fraction of samples in which it is present?

RESPONSE: Yes. It has ben reformulated now.

L342-343: What is “occurrence frequency”?

RESPONSE: has been changed to simply “occurrence“ (=the number of

samples which the OTU is observed in) with an added explanation in parentheses.

“Plant Survey Data”: The broad range of environments included here is impressive.

L456-458: How were non-plant OTUs discarded?

RESPONSE: This is explained under the heading “Taxonomic assignment and filtering”. A cross reference has been added.

L506: When citing a web page, it is recommended to date it (especially for version-controlled web sites like this one, as then the exact same page can be retrieved later on).

RESPONSE: An accession date has been added.

L514-515: The DADA2 method does correct PCR errors.

RESPONSE: We deleted our postulate/assumption, as it is not so important. (a note: how can DADA2 identify simple errors arising during PCR? If occurring in the first rounds of PCR it will be indistinguishable from true biological variants, as it will occur in high abundance and have good sequence quality. Am I wrong?)

Table 1: If space permits, I would like to see the slope/intercept values included in this table as well.

RESPONSE: Good idea. Now included in the table.

Supp. Figure 1a/b: Need a high-resolution version of these figures. Can't read any of the fit parameters, which I'm quite interested to see.

RESPONSE: We are very sorry for this. We have now submitted high-res versions, and included the regression statistics in the table.

Reviewer #2 (Remarks to the Author):

In this paper they have developed a post-clustering curation method of pre-processed metagenome sequencing data by merging erroneous OTUs into a major cluster. They have validated this method using plant metagenome data and improved the accuracy of the richness estimates. However, algorithmic details are unclear and support of their hypothesis is validated in only special case which is not generalizable. Substantial revision is needed to be acceptable

in the journal.

General comments

1. The authors stated that the algorithm does not require arbitrary cut-off levels. However, some user-selected default parameters exist in the algorithm. The author must clearly explain this statement and discuss the advantage over previous methods of applying arbitrary level to discard erroneous OTUs.

RESPONSE: We cite a study showing that singleton culling will throw real biological data out. We have now included a full analysis of the effect of culling singletons in all datasets. Although we cannot unambiguously evaluate whether we lose real species in that process, we can test whether this approach results in improved metrics, which it only does to a minor degree. This has now been included in the manuscript. Also we have included a visualization of the abundance of errors (taxonomically redundant at least) before and after curation for the OTUs kept and discarded in the different approaches for selected genera of plants. Here it is evident that many errors are abundant, and that no universal cut-off will eliminate these without simultaneously throwing real biology out. This has been included in the manuscript and as supplementary figures.

2. It is unclear to me that this present method is the only curation method available for amplicon sequencing data. There exists a paper (doi:10.1128/AEM.01043-13) using dual indexing strategy. The authors should discuss what has been done before and compare the strategies.

RESPONSE: Of course. We have now included more information on what has been done, both to reduce the number of errors formed (e.g. dual indexing), and removing errors (e.g. chimera tools). Our PCR setup is strictly dual-indexing, and we do not re-use any tag in any library, and thus chimera formation (between samples) should not be an issue as we are not looking for the chimeric tag-combinations. Our initial processing with VSEARCH includes a chimera removal step, and in the DADA2 pipeline we also employed the implemented tool for chimera removal. These things have now been written more clearly. And these approaches for error-minimization and removal have been included more thoroughly in the text now.

3. In comparing total OTU richness versus total survey species richness, the CROP algorithm was the only method to underestimate the total richness and did not improve much after curation. It would be interesting to discuss what drives this result.

RESPONSE: We tried to investigate what was causing the suboptimal behavior of the algorithm without success, despite re-runs. The processing itself takes many days, and we could not investigate further. It seems that it

in reality uses a broader OTU definition than what is indicated in the manual, but not least importantly, it selects representative sequences for the OTUs that are not the most abundant (or likely) ones, often, which for example results in the selected sequence for the *Fagus sylvatica* (our most abundant OTU/species) to be a sequence with a best match on GenBank of 95% (whereas the other algorithm all retain the abundant haplotype with 100% match as reference sequence). I am sorry that we cannot elaborate on this. However, we have now included detailed plots showing the OTUs identified (and retained/discarded by LULU) for selected plant genera as supplementary information. We hope this will suffice.

4. The authors insist that LULU algorithm greatly improve the estimates of the accuracy of richness derived from amplicon sequencing data in various measures such as alpha or beta diversity. Their algorithm was developed based on well-studied organism. It would be interesting to know if this post-clustering algorithm applies well to the other organism groups. Further evaluation at other application would be helpful to validate the results.

RESPONSE: The reason why we chose to validate the algorithm on vascular plants is that it is the only group of organisms where we can produce a real, natural ground truth dataset for comparison/validation. We wanted to validate the algorithm of sequences produced from real, complex environmental samples, and not on mock communities without the complexity of DNA from non-target organisms, inhibitors, etc. Validation on any other organism group from soil DNA (fungi, nematodes, earth worms, etc) would be from improvement of a few secondary indicators (e.g. improved average match to genbank), or require specialized methods for extraction of the organisms and substantial taxonomic expertise. As the algorithm is entirely independent of taxonomic annotation of OTUs, it works equally well on any other organism group with a genetic structure comparable to ITS in plants. Of course we have tested it on other organism groups, and judging from the results, it performs well. Only, we cannot substantiate these results by more than indications and have refrained from including these, as conclusions tend to be speculative.

5. In algorithm section, description is too vague to understand. Although they said any sequence comparison tool can be used for measuring similarities, the tool they have used in this study needs to be specified. In addition, the process of making match list is not properly described how focal OTUs can be found in this process and how to find potential parent OTUs. Even supplement note does not have information for generating this list. Furthermore, this section seems to have mixed information of description of how to use this LULU and LULU's analysis step. It will be better if authors separate these two in different section or upload manual on github page.

RESPONSE: We have now tried to disentangle the section on the algorithm.

In the manuscript we use BLASTn to produce the match lists. The exact commands have now been included in the methods section. The methods section includes many scripts that were not submitted as part of the initial submission available to reviewers (including the script that makes the match lists). All scripts used will be uploaded on GitHub, and included as supplementary material (if possible), and can of course be made available to reviewers upon request before then. And all information necessary to install and run the algorithm is available on GitHub. Also we have added a figure describing the workflow.

Minor comments

1. The term “OTU” needs to be defined at the first time.

RESPONSE: Yes. Done

2. All the supplementary figures lack resolution that was hardly recognizable.

RESPONSE: Sorry, resubmission contains high-resolution figures.

3. There is only one figure summarizing the result. The paper focuses on the algorithm and needs the schematic figure flow of their core logic. In the methods section, the algorithm is written too general and it was hard to know the distinctive advantage over other methods. It needs to be clarified and be more specific to the readers.

RESPONSE: We agree. The algorithm is now explained more detailed, and a figure describing the workflow is included. More detailed instructions for use can also be found on GitHub. We considered including some of the supplementary figures in the main text, but decided that it maybe would make the intended information less obvious.

4. For the most, LULU’s github page mentioned in this manuscript is not available currently and needs to be fixed.

RESPONSE: It is now open and available.

5. There are some minor typos throughout the paper.

RESPONSE: We have tried to catch them now.

Reviewer #3 (Remarks to the Author):

Frušlev et al. present an algorithm called ‘LULU’ that attempts to reduce impacts of DNA sequencing errors on estimates of species richness in DNA metabarcoding surveys. Relying on assumptions about the signatures of DNA

sequencing errors in these datasets, the algorithm is shown to reconcile estimates of species richness obtained from field survey data of vascular plants with the richness of plant DNA in soil cores at the same sites. The authors interpret this as evidence that errant DNA sequences have been culled from the dataset.

The problem of accurately estimating species richness from sequence-based surveys is widespread and needs to be tackled. I commend the authors for addressing this issue in a comparative approach, but I have several questions/concerns that still need to be addressed so that the reader can gauge the authors' assumptions/interpretations in comparison to the alternatives.

General comments:

1. In “validating” the algorithm, the authors suggest “true” sequences are retained and “false” sequences are discarded. Results are consistent with this suggestion, but this cannot definitively be determined from correlations with plant surveys; experiments using “mock communities” of known composition would be required.

RESPONSE: Yes, we agree, that a true validation would require samples with known content - i.e. mock communities. However, that is a completely separate study, which we of course would applaud. Despite the obvious advantages of mock communities, it is difficult to achieve the complexity of real samples with DNA from all sorts of non-target organisms, which we assume causes more errors in real samples. We are confident that the improved metrics can be taken as a strong indication of validity, as random removal of OTUs would generally not improve metrics (e.g. r-square values of OTU richness vs. species richness).

2. The second half of the introduction focuses heavily on methods but lacks detail, leaving the reader to interpolate how the main results of the manuscript relates to the broader problem being addressed. See several comments below.

3. The authors are critical of several previously used strategies to “curate” similar datasets—such as arbitrary decisions to cull singletons from OTU tables—and rightfully so. However, they did report how the inferred level of species richness based on these decisions differ from the output of LULU; instead, they reported how a set of uncurated OTU tables compare to the output of LULU. There are two potential issues:

(a) Without experimentally classifying OTUs as “true” or “false” (e.g., from mock communities), any method that reduces a bad overestimate of richness to a more reasonable value—including arbitrary culling of singletons or inaccurate culling of some “true” sequences—can appear to improve correspondence with ecological surveys. Thus, the reported correlations may be suggestive of increased accuracy after applying LULU, but causation remains unclear.

RESPONSE: We agree that our validations are indications, but based on comparison with thorough “ground truth data”. And we have now added the need for mock community tests to the manuscript, with the reservations we have for this approach. Random removal of OTUs will of course lower the overestimates of OTU richness, and in most instances also the beta-diversity and taxonomic redundancy. But we insist that random culling of OTUs will not cause better correlations (r -square value) of OTU richness vs. observed plant richness. It will cause a lower slope, and possibly an intercept closer to zero, but not improve the fit. Neither will random culling cause improved community dissimilarity metrics, lowered proportion of taxonomically redundant OTUs and increased average reference database match. We have now included tests where we exclude singletons during the processing to test and illustrate this.

(b) Most researchers go beyond clustering OTUs in their attempts to remove errant sequences from DNA metabarcoding data. Therefore, it would be better to compare the output of LULU to both the raw OTU tables and to several commonly employed sets of assumptions (e.g., culling singletons) and/or existing algorithms designed to remove errors and denoise these types of datasets. By comparing LULU’s output to OTU tables without any additional effort to remove errors, the authors may inadvertently present a false dichotomy.

RESPONSE: Actually, we do include denoising (using quality cut off and chimera removal) where possible (initial processing, VSEARCH, DADA2), and used safe quality cut-offs (sequences with expected error rate above 0.002 or containing 1 or more N’s were discarded). This has now been written more clearly in the text. Along with the comments to the issues above (test with culling of singletons), we hope that it is satisfactory.

Title: Is “post-clustering” a very searchable/informative term to use?

RESPONSE: We have now added the term ‘co-occurrence based’. We think that co-occurrence based post-clustering curation is an adequate term, but are open to alternative suggestions.

Line 15: I suggest using “DNA metabarcoding” to match the lexicon.

RESPONSE: Agreed

Line 16: Is “LULU” an acronym that should be defined?

RESPONSE: no it is the name of first authors daughter. A bit silly, but it stuck during the process.

Line 19: Be careful to provide context for the word “barcode”. There are

differences in implied meaning between research communities, as some refer to the standard markers used by the “barcode of life” (e.g., COI) while others refer to molecular identification tags (i.e., “MID-tags”).

RESPONSE: Agree. I removed the ambiguous word and simply use “community metabarcoding data from amplified marker genes”

Line 20: Is a minimum number of samples needed to make strong inferences based on co-occurrence patterns? The abstract may not be the appropriate place to address this question, but it needs to be addressed clearly.

RESPONSE: This is a difficult question to address without numerous and complex mock communities, and it is highly context dependent. Low complexity and similar communities containing genetically closely related species will be subject to erroneously removed OTUs by LULU curation when analyzing few samples only, whereas highly complex communities with less related species, will perform better. Renalyses of our data with culling of samples/sites would be indicative, but these analysis will be highly time consuming only indicative in the conclusions.

Line 20-21: Specifically, what types of markers, organisms, or environmental gradients were sampled?

RESPONSE: Now also included in abstract.

Lines 25-27: Although LULU may not require a reference database to run, please be careful to avoid implying that reference databases are not needed/useful for assessing the accuracy or completeness of DNA metabarcoding studies. Indeed, your own assessments employ one.

RESPONSE: good point! Has added a line in final sentence of the discussion!

Line 35: “poorly studied” – would it be more constructive to describe the specific ways in which previous research has not completely addressed this issue?

RESPONSE: A little addition to that sentence has been made. Otherwise, I believe we are addressing this somewhat elsewhere in the text. Otherwise give me a hint on which “specific ways” you are hinting at.

Line 53: “many spurious OTUs” – could the results of this reference be presented more precisely, and perhaps in comparison to the “38%” estimated in the previous reference?

RESPONSE: An extended sentence added to explain that the main source of erroneous OTUs are due to the greedy nature of the algorithms making

them incapable of identifying low abundance variants that were missed in the clustering.

Lines 56-58: Although it is true that researchers do this, it is also true that these decisions should depend on the marker and taxonomic group they employ.

RESPONSE: a clarifying remark added.

Lines 58-65: Good – this concisely captures a major problem: the ecological reality of rare species needs to be accounted for in algorithmic approaches to curating datasets. Please also note that the presence of DNA in a substance (e.g., soil) does not imply that it corresponds to an organism with any biological function in that substance. Thus, there may be “true” DNA sequences in a sample that should not be targeted for removal by algorithms but that also may lack any relationship with other ecological data (e.g., local plant surveys).

RESPONSE: Agreed. We have a general comment circumscribing issues like this in the introduction “Not only are there sampling issues with regard to environmental DNA”, but believe it is out of scope to dive into the many false positives, positive negatives, and negative negatives in a this kind of study in general, as these have been adequately covered in the many reviews published on general issues of “HTS metabarcoding”

Lines 77-84: Much more detail on the surveys are needed: where in the world were these surveys; how were they sampled; what are the expected levels or ranges of richness based on these surveys; across which taxa are these ITS markers “comprehensive” in coverage; what is the nature of the “reference data” for vascular plants (i.e., presence vs. absence or abundance)? Although lengthy details can be saved for the methods, the reader will struggle to understand the results as written.

RESPONSE: Agreed. Detail has now been added. All the remaining information should be available in the methods section. We have included an assessment of the availability of target sequence (ITS2) of species identified in the plant survey, and added this to the manuscript. As the taxonomic annotation of reference database sequences is only used at generic level, and to exclude non-plant OTUs from the comparisons, we do not see the need to make a detailed evaluation of the more precise annotation quality and completeness of GenBank.

Lines 82-84: By what criteria were GenBank data deemed “adequately populated”? Were all of the species recovered in the plant surveys also present in the database? If not, how few?

RESPONSE: We have now done analyses on this, and included it in the manuscript.

Line 86: Do the terms “curated” and “un-curated” refer to datasets that have been subjected to the LULU algorithm or not, respectively? I suggest stating this clearly, and being consistent throughout the text.

RESPONSE: Good point. We have tried to homogenize this now.

Lines 86-97: This paragraph is methods-heavy—indeed sections of it seem to be lifted from text that also occurs within the methods section. I suggest making this the first paragraph of the “Results” section, and using it to define the structure of the results section, in which multiple comparisons are presented sequentially under subheadings (even though all of these subsections refer to Figure 1).

RESPONSE: Agree. We have deleted that section (as it was almost identical to the corresponding section in the methods section. The last part of the introduction now functions as a bridge to the results.

Lines 95-96: Please give a concise definition of “Taxonomic redundancy” and “distribution of reference database matches” to save the reader from having to flip through the methods section.

RESPONSE: Good point. We have added a short description for most of the metrics in that section now.

Line 108: assumption #3 – this assumption needs to be supported rigorously, or perhaps reconsidered, because opposite expectation could also be supported. For example, (a) soil particles can bind and preserve plant DNA long after local plant composition changes (~years to centuries), (b) soil can contain DNA of plants that are not locally apparent (e.g., from pollen, dormancy, etc.), (c) unique ITS sequences within the soil may (truly) be more numerous than the plant species that contributed to them, due to the concerted evolution and intragenomic variation that can plague ITS-based studies. While the presented expectation may hold under some circumstances, this is likely to be scale-dependent and the reader does not know the scale of your sampling regimes at this stage.

RESPONSE: We agree that this potential of “non-active” plant DNA is present in the soil and may pose a problem when assessing the active community. We have added a passage on this and related it to our sampling scheme and the selected marker region in the methods section, and referred to this in the results section (where indicated). We assume that the length of the amplicon along with the relative productive soils of our region result in this non-active DNA not being amplified to a great extent. In combination with the fact that we only sampled 0.01% of the surface of the survey area, we feel it safe to assume that the DNA extracts will contain DNA from less species than the survey finds. (Also we do not

see a major contribution of OTUs - species names of these - that doesn't fit our expectation based on our knowledge of the sites. But this is out of scope for this paper)

Line 131: Good – the application of LULU may not only “improve” the selected measures, but in 3 / 4 of comparisons the resulting counts fell below your ‘maximum’ criterion of 564 species. This could be stated more directly.

RESPONSE: This has now been formulated more precisely. Actually the total OTU count is below 564 in all of the curated tables.

Line 141: I suggest trying to justify this approach with citations and/or by presenting a separate analysis of the available GenBank data.

RESPONSE: This has now been substantiated with an analysis of GenBank data. Paragraph has been rephrased, and the additional analyses have been added to the corresponding methods section.

Line 159: “simple beta-diversity measure” – which one? Was it a weighted or unweighted metric?

RESPONSE: A short description has been added in that paragraph.

Line 169: “Assuming that Genbank is well populated” – this should be easy to check by directly comparing vegetation survey data to GenBank data.

RESPONSE: This analysis has now been added, and the related paragraphs rewritten.

Lines 313-314: In what format? Does this format match an output that can be obtained directly from each of the OTU-clustering programs utilized?

RESPONSE: Paragraph has been made more precise. It is a simple tab delimited file. The instructions on GitHub should be OK for more details on use by readers.

Line 339: Seems like there is a word missing or out of place in this sentence.

RESPONSE: The wording of that and adjoining sections has been adjusted.

Line 393: What is known about the taxonomic “coverage” and species-level “specificity” of these markers? (See, e.g., Ficetola et al. 2010 for discussion of coverage and specificity)

RESPONSE: Good point. Reference to studies evaluating the ITS2 marker for plants has now been added.

Line 410: subscript the 2 in “MgCl₂”

RESPONSE: done

Lines 439-440 (and elsewhere; 505-506): If one goal of the manuscript is to provide a method for wide use by the research community, I suggest minimizing “custom” scripts in your methods or at least providing relevant references to published/commercial strategies for accomplishing the same task.

RESPONSE: Good idea. I have tried to make it more clear, that the LULU algorithm can be used on any OTU table produced with any clustering tool. The focus of the paper is the application of the LULU method, and I hope that is clear. This method (LULU) requires no custom scripts (except for the method itself). The initial processing of raw read data and the formation of the OTU tables can be done with any established pipeline, and we do not want to make further recommendations for that selection than we already do(in the discussion). The DADA2 processing we apply is based on the published processing with that tool with specific adjustment suggested by the authors for accommodating our tagging approach (also online). At the time we initiated our analyses, I could find no demultiplexing scripts allowing for the use of dual tags with different lengths. It may be published now, but I have no experience with these potential alternatives, and cannot make recommendations. Again, the main focus of this manuscript is to present an OTU table curation tool, which should be easy to use, and only requires an OTU table, the sequences, and a little knowledge of R and commandline tools. (Also, I would not like to give the impression that these kinds of analyses in general can be done with a simple plug-and-play solution ☺)

Lines 454-456: Analyzing data available in GenBank could also test this assumption. For plant species known from the plant surveys, is this assumption upheld consistently? If too few of the local species have representative ITS sequences in GenBank, this should be acknowledged, but the specificity of the marker could still be evaluated (e.g., using “ecoPCR” as in Ficetola et al., 2010).

RESPONSE: The problem of using ecoPCR on ITS data from GenBank is that almost all GenBank sequence have had the primer regions excised, and thus will not be retained/identified by ecoPCR. We have now added references to studies examining the intra- and interspecific variation of ITS2 for plants, and hope that this will suffice, as these studies are far more elaborate than anything we can include here. Also we did a search on GenBank to check the availability of ITS2 sequences for the species identified in our survey, and included this in the manuscript.

Line 530: “0.51%” – is this value meant to be a percent or a proportion?

RESPONSE: Rewritten

Lines 533-536: What proportion of sequence reads and unique sequences were retained within the array of samples?

RESPONSE: This has now been added.

Lines 580-582: Also note the potential for intragenomic variation using ITS.

RESPONSE: Now included in the paragraph.

Lines 614-616: “We postulate...removal of OTUs cannot result in... increased dissimilarity” – why not?

RESPONSE: There was a mistake in the formulation. It has been reformulated and extended. We intended to postulate that removal of erroneous OTUs could not lead to increased genus level dissimilarity of the communities on average, under the assumption that we are actually sampling the same communities with the survey and the eDNA approach, etc.

Lines 618-620: “An evaluation of community composition relies heavily on correct annotation” – this is not necessarily true as written. It is certainly possible to compare the composition of two species x site matrices in the complete absence of Latin binomials. I would contend that the estimated dissimilarity of sites based on a standard metric (e.g., Bray-Curtis) would be highly concordant between comparisons based on curated and uncurated OTU tables. This conjecture could be evaluated using Mantel tests. I suggest the authors present such analyses because, at the very least, it brings into consideration whether the putative sequencing errors that LULU was developed to remove might impact estimates of species richness (the focus of the manuscript’s introduction) as well as ecological dissimilarity (which is likely less sensitive to these sequencing errors).

RESPONSE: Oh, thanks. We were actually talking about the TAXONOMIC composition – i.e. correspondence in the taxonomic annotation of the OTUs and the corresponding identified plants for the single sites. We have now reformulated the whole paragraph and used the more adequate term “taxonomic dissimilarity”. We have also added the suggested analyses of community composition - both for pre and post-curation tables and for OTU vs. plant community.

Line 656: “trend” vs. “trends”

RESPONSE: The title of that paper (<http://dx.doi.org/10.1016/j.tree.2014.11.006>) is actually “Fifteen forms of

□ biodiversity trend in the Anthropocene” 😊 □

Reviewers' comments:

Reviewer #1 (Remarks to the Author):

I have reviewed the revisions made by Frøslev et al, as well as their responses to both my comments and those of the other reviewers, and find them acceptable and now recommend publication. In particular, the critical flaw of overlooking the distribution-based clustering method has been completely rectified, and I believe the addition of comparisons with that method, as well as the removal of singletons has made the paper stronger. I applaud the reporting of results from dbOTU3 as both a primary clustering method, and as a post-curation method.

Three minor comments below:

L25: I recommend clarifying "Results" here -- as I read the paper the results were not identical between initial algorithms, but rather LULU improved the reported quality metrics for all initial algorithms.

L56-59: This added sentence needs to be revised. There are multiple sources of low-abundance errors, and certainly not all can be ascribed to the greedy nature of the most common algorithms. That is a factor, not the factor.

L203: Missing text?

Reviewer #2 (Remarks to the Author):

The authors have done a lot of work addressing the comments for three reviews. I commend the authors for providing workflow of algorithm to make reader easier to interpret LULU since the manuscript heavily relies on explaining algorithm. But, I had hard time catching up the answers in the reviewer's responses. The authors should reply with the page and line number to indicate what they have modified in the original manuscript. I have a few more minor comments for improving the ambiguous interpretations. After answering all the issues properly, the manuscript is suitable for the publication.

Minor comments:

Although, the term "co-occurrence" and the "taxonomic redundancy" are explained in the methods section, it would be good to specify in the main text so as readers to understand the concept clearly. On line 356, "Traditional removal of singletons~", what extent does the traditional removal of singletons are less efficient than LULU? Please provide the specific value for clarity. Please provide any pre-requisites of LULU algorithm on the main screen window as they have described under Files_LULU_manuscript folder. Also, at least in github page, it would be nice if they show the computing resource usage guidelines

Reviewer #3 (Remarks to the Author):

The manuscript has improved substantially and it is easier to appreciate the authors' belief that the LULU algorithm constitutes a substantial improvement. A few areas remain where I suspect that clarity/transparency could be improved, but I also suspect that many readers will want to compare this algorithm to whatever approach they are currently using to curate similar datasets.

Two general and interrelated comments:

1. I remain skeptical of criterion #3 (lines 133-136), that a soil-based eDNA survey should yield lower (true) OTU richness than a local botanical survey. There are several reasons: (1) soil binds and stabilizes eDNA for very long periods of time and thus may include taxa that are no longer present, (2) the DNA of non-local plants may occur in soil (e.g., from pollen, litter), (3) the DNA of cryptic plants (e.g., dormant plants, trailing roots) may be sampled, (4) cryptic intraspecific and intragenomic DNA variation may exist generate multiple OTUs per species that would not (could not) be recorded by botanists in the field. All of these possibilities (and possibly others that can be found in the literature) would lead to the observation of greater OTU richness in every plot than would be recorded in field surveys. If the authors should more rigorously support this expectation and at least acknowledge the support that would exist for alternative expectations.

2. The authors' concern about overestimates of OTU richness are valid and important to address, and they make a strong case that LULU improves estimates of richness, but support for the claim that LULU is selecting the "correct" OTUs is still weaker than it should be. This also gets to the crux of my concerns about criterion #3 (mentioned above). Are the selected OTUs a smaller subset of the plant species known to be present in a plot (i.e., a fully nested subset), can some proportion of the OTUs be attributed to taxa that are not known to be present in a plot, and/or are some OTUs taxonomically redundant? With the availability of reference DNA attributed 88% of plant taxa recorded in the field surveys, it should be possible to answer this question more directly than it was apparent to me in the current version of the manuscript. After reducing OTU richness below the 1:1 line, what proportion of OTUs or sequences reads definitively matched species recorded in the survey, what proportion definitively did not, and what proportion could not be classified either way (e.g., because they lack a match in GenBank and remain poorly identified)? This is related to the information presented in the "Taxonomic Dissimilarity" – but more directly informative. Inferences from the Taxonomic Dissimilarity section are a bit more complicated, because comparisons between DNA- and field-based site composition data are presented via the presumed degree of improvement made by LULU over each of the original algorithms, but not with reference to the a priori expectation of OTUs that should be found in each sample (unless I am not clearly understanding the text).

More specific comments:

Line 14: "monitoring" – of what?

Line 21: "comprised a high quality survey" – drop the "a"?

Line 32: "marker genes" – these are not necessarily gene regions, are they?

Line 37: "ground truth and DNA data" – the expression 'ground truth data' is awkward. Rephrase?

Line 100: "Comprehensive" – not quite the right definition; perhaps "extensive"?

Line 150: "significantly" – If the result of a statistical test, please provide results. If qualitative, please state the magnitude of the difference (e.g., two-fold?).

Lines 154-156: I see in response to previous reviews that CROP algorithm behaves in different ways and the authors are uncertain how to explain it. Can it be stated in the main text what is unknown (or what should be known) about the algorithm?

Line 203: This sentence ends abruptly. Where should we look to visualize this?

Line 205 (captions for Fig. 1f and Fig. S6): In the Taxonomic Dissimilarity section, the choice and

interpretation of the dissimilarity metric could be explained more clearly. As written, it is hard to know what a "good" result would look like. Would a "perfect" result be taxonomic dissimilarity of zero for each plot? This gets a bit confusing, in part because the result is expressed as a before/after correlation and the eye has a tendency to want to see a strong correlation in Figs. 1f and S6. For this reason, I had to read through the text and captions several times to realize that the best result was not CROP. It is explained that the goal is to pull points to the "right" of the 1:1 line, but really the goal is to pull points close to 0 on the y-axis, regardless of where they start (left to right) along the x-axis. Once this is established, it does become clear that LULU improves the results of some algorithms without improving the results of others; nevertheless, there is variation among starting conditions based on the output of each algorithm (indeed, some do not require as much improvement, as indicated by their truncated distribution along the x-axis). Thus, while it is true that CROP is not improved as much as some of the others (e.g., SWARM), it is also true that CROP did not require as much improvement (at least according to this metric). What also appears true, is that a considerable amount of unexplained taxonomic dissimilarity remains following the application of LULU: far from having a taxonomic dissimilarity of 0, results range from ~0-100 across sites and algorithms. This very definitely must be acknowledged in the main text – it is promising that LULU shows improvement in some algorithms, but a more simple presentation (e.g., histogram) showing the final distribution of taxonomic dissimilarities would make it clear that these two datasets still do not correspond. (Again, whether or not they SHOULD perfectly correspond is another issue.)

Line 219: "Community composition is mainly driven" – not the composition of the community itself, just the metric that is calculated.

Lines 238-239: From the methods section it is clear that field-based matrices were not weighted by any measure of abundance. That should be made clear here as well, and it should be considered whether accounting for relative read abundances (vs. presence/absence) in the DNA matrix is sensible in this case.

Lines 328: "alpha-diversity" – and "alpha-composition"? Both should be addressed.

Line 360: "adequate for studies of community dissimilarity" – but perhaps not for studies of community composition (i.e., the identity AND richness of species)?

Lines 373-374: "CROP was the method least receptive..." – Again, is this because CROP started off in a "good" place, or is this lack of receptivity problematic in some way?

Reviewers' comments:

Reviewer #1 (Remarks to the Author):

I have reviewed the revisions made by Frøslev et al, as well as their responses to both my comments and those of the other reviewers, and find them acceptable and now recommend publication. In particular, the critical flaw of overlooking the distribution-based clustering method has been completely rectified, and I believe the addition of comparisons with that method, as well as the removal of singletons has made the paper stronger. I applaud the reporting of results from dbOTU3 as both a primary clustering method, and as a post-curation method.

Three minor comments below:

L25: I recommend clarifying "Results" here -- as I read the paper the results were not identical between initial algorithms, but rather LULU improved the reported quality metrics for all initial algorithms.

RESPONSE: Rewritten. Now reads (L23-L26):

“OTU tables were produced with several different OTU definition algorithms and subsequently curated with LULU, and validated against field survey data. Curation consistently improves α -diversity estimates and other biodiversity metrics.”

L56-59: This added sentence needs to be revised. There are multiple sources of low-abundance errors, and certainly not all can be ascribed to the greedy nature of the most common algorithms. That is _a_ factor, not _the_ factor.

RESPONSE: Sentence deleted (a little out of context, anyway).

L203: Missing text?

RESPONSE: Deleted. (It was a remnant of an earlier version where the section “Curation effect on selected plant genera” was build into this section)

Reviewer #2 (Remarks to the Author):

The authors have done a lot of work addressing the comments for three reviews. I commend the authors for providing workflow of algorithm to make reader easier to interpret LULU since the manuscript heavily relies on explaining algorithm. But, I had hard time catching up the answers in the reviewer's responses. The authors should reply with the page and line number to indicate what they have modified in the original manuscript. I have a few more minor comments for improving the ambiguous interpretations. After answering all the issues properly, the manuscript is suitable for the publication.

RESPONSE: We are sorry. We submitted a word document with fully highlighted changes, as we thought you would be able to download this. In this revision we have made a simple highlighting in yellow of the changed text, and also included more information on the changes here in the response-file. We hope this is satisfactory.

Minor comments:

Although, the term “co-occurrence” and the “taxonomic redundancy” are explained in the methods section, it would be good to specify in the main text so as readers to understand the concept clearly.

RESPONSE: co-occurrence is explained in the introduction with the sentence (L80):

“The core mechanism is the identification and merging of ‘daughter’ OTUs with consistently co-occurring, sequence-similar, but more abundant ‘parent’ OTUs across a multi-sample dataset, under the assumption that the ‘daughter’ OTUs are artefacts.”

Does this need to be elaborated further?

A sentence on the metric “taxonomic redundancy” has been added now in that result section (L168):

“(proportion of OTUs with a taxonomic annotation already represented by another OTU in the table).”

On line 356, “Traditional removal of singletons~”, what extent does the traditional removal of singletons are less efficient than LULU? Please provide the specific value for clarity.

RESPONSE: All the singleton removal metrics are given in Supplementary Figs. 7-11 and Supplementary Tables 3-4 as written in the results section “Singleton removal compared to post-clustering curation”. We have reworded the corresponding results section to include a comparison of the percentage of improvement of the correlation of OTU and plant richness, compared to LULU curation, and it now reads (L264):

“Singleton removal had some positive impact on several measures, especially for the approaches using greedy algorithms and low clustering levels. But no metrics were improved to a degree similar to LULU curation, e.g. the coefficients of determination (R^2) for the correlation between OTU richness and plant richness were improved - 0.01 to 0.11 (mean improvement 0.03) by singleton removal, compared to the mean improvement of 0.27 with LULU curation”.

And we also added this calculation (percentage improvement) to the section “Curation improves correlation with plant richness” (L138).

Please provide any pre-requisites of LULU algorithm on the main screen window as they have described under Files_LULU_manuscript folder. Also, at least in github page, it would be nice if they show the computing resource usage guidelines

RESPONSE: Prerequisites have been added to the main github page.

There is already a step-by-step usage explanation, and I am unsure what is meant by “show the computing resource usage guidelines”.

Reviewer #3 (Remarks to the Author):

The manuscript has improved substantially and it is easier to appreciate the authors' belief that the LULU algorithm constitutes a substantial improvement. A few areas remain where I suspect that clarity/transparency could be improved, but I also suspect that many readers will want to compare this algorithm to whatever approach they are currently using to curate similar datasets.

Two general and interrelated comments:

1. I remain skeptical of criterion #3 (lines 133-136), that a soil-based eDNA survey should yield lower (true) OTU richness than a local botanical survey. There are several reasons: (1) soil binds and stabilizes eDNA for very long periods of time and thus may include taxa that are no longer present, (2) the DNA of non-local plants may occur in soil (e.g., from pollen, litter), (3) the DNA of cryptic plants (e.g., dormant plants, trailing roots) may be sampled, (4) cryptic intraspecific and intragenomic DNA variation may exist generate multiple OTUs per species that would not (could not) be recorded by botanists in the field. All of these possibilities (and possibly others that can be found in the literature) would lead to the observation of greater OTU richness in every plot than would be recorded in field surveys. If the authors should more rigorously support this expectation and at least acknowledge the support that would exist for alternative expectations.

RESPONSE: It is important observations and should of course not be reserved to the methods section. We have now extended the wording of this results section, and it now reads (L133):

“...although DNA from soil for several reasons may in fact have DNA from more species than are apparent at the time of investigation (see methods section).”

We have further extended the paragraph in the methods section (that is referred to) explaining our assumptions in more depth. In full, it now reads (L773-L790):

“DNA extracted from soil may contain DNA from more plant species than can be observed in a classical survey. Soil may harbour DNA from plants that are not apparent or biologically active at the time of investigation as these may be represented by e.g. pollen, seeds, etc, and soil particles can bind and preserve DNA from species no longer present. Furthermore, morphologically defined species may harbour cryptic but genetically separate species, and also the investigated ITS2 region is known to have varying levels of intraspecific and intragenomic variation that is difficult to accommodate within an universal clustering level. However, the sampling of this study was carried out in a temperate environment with productive soils, and we assume that these DNA remains will be present in too low abundance and too fragmented to

amplified with the selected primers (targeting a region of 300+ bp on average) to pose a problematic contribution to the sequence pool. Considering this and taking into account, that the soil sampling only covers approximately 0.01% of the soil surface of the 40 m × 40 m sites, we expect the sequencing approach to underestimate the true diversity, because many species occurring with few or small individuals only are likely to be missed”.

2. The authors' concern about overestimates of OTU richness are valid and important to address, and they make a strong case that LULU improves estimates of richness, but support for the claim that LULU is selecting the “correct” OTUs is still weaker than it should be. This also gets to the crux of my concerns about criterion #3 (mentioned above). Are the selected OTUs a smaller subset of the plant species known to be present in a plot (i.e., a fully nested subset), can some proportion of the OTUs be attributed to taxa that are not known to be present in a plot, and/or are some OTUs taxonomically redundant? With the availability of reference DNA attributed 88% of plant taxa recorded in the field surveys, it should be possible to answer this question more directly than it was apparent to me in the current version of the manuscript. After reducing OTU richness below the 1:1 line, what proportion of OTUs or sequences reads definitively matched species recorded in the survey, what proportion definitively did not, and what proportion could not be classified either way (e.g., because they lack a match in GenBank and remain poorly identified)? This is related to the information presented in the “Taxonomic Dissimilarity” – but more directly informative. Inferences from the Taxonomic Dissimilarity section are a bit more complicated, because comparisons between DNA- and field-based site composition data are presented via the presumed degree of improvement made by LULU over each of the original algorithms, but not with reference to the a priori expectation of OTUs that should be found in each sample (unless I am not clearly understanding the text).

RESPONSE: This is a good observation. We initially tried to stay away from analyses depending on exact interpretations of the reference database annotation, and to keep analyses at the “method level” except for the pairwise comparisons of richness and the genus level “taxonomic dissimilarity metric”. But, to qualify that LULU actually improves the “alpha-composition”, we need these more precise species level comparisons, you are right. We have now done a precise comparison of the species-level, site-wise taxonomic composition of each site and calculated the metrics suggested. This is all added as a new section: “Taxonomic composition” (L206-L225 and L859-L886) and the new table (Table 2). We also calculated the proportion of recaptured species being lost during curation, and added this along with a discussion of this in the discussion section (L377-L381). These calculation/comparison are based on a comparison of OTU annotation with plant data from a slightly extended survey dataset including observational data from two supplemental surveys to include as many plant species as possible (for classifying recaptured/unregistered

species). These supplemental data were however not collected in a standardized way, which is why they were not (and still are not) used for the richness comparisons. This information has been added to the methods section. The new “Taxonomic composition” section makes the difficult-to-interpret section “Taxonomic dissimilarity” irrelevant, as it was - as you observe - basically a less direct evaluation of the same thing. Thus, it has now been deleted (and replaced by “Taxonomic composition”.)

More specific comments:

Line 14: “monitoring” – of what?

RESPONSE: now (L14):

“...cost-effective biodiversity monitoring”

Line 21: “comprised a high quality survey” – drop the “a”?

RESPONSE: dropped

Line 32: “marker genes” – these are not necessarily gene regions, are they?

RESPONSE: now reads (L32):

“..genetic markers..”

Line 37: “ground truth and DNA data” – the expression ‘ground truth data’ is awkward. Rephrase?

RESPONSE: Now reads (L37):

“...paired sets of thorough inventory data and DNA data”

Line 100: “Comprehensive” – not quite the right definition; perhaps “extensive”?

RESPONSE: Now reads (L96):

“...from an extensive soil sampling...”

Line 150: “significantly” – If the result of a statistical test, please provide results. If qualitative, please state the magnitude of the difference (e.g., two-fold?).

RESPONSE: Now reads (L151-L152):

“...identifying 2.3 to 27 times more OTUs (1,320 to 14,828) than observed plant species at all clustering levels”

Lines 154-156: I see in response to previous reviews that CROP algorithm behaves in different ways and the authors are uncertain how to explain it. Can it be stated in the main text what is unknown (or what should be known) about the algorithm?

RESPONSE: We have now written more clearly that the main problem is that it retains suboptimal representative sequences of the retained OTUs. I am unsure if we can explain more. The CROP algorithm retains fewer OTUs, and the representative sequences are not ideal (i.e. they have a lower average match with reference data), and the taxonomic redundancy is still high. All these things can be read/seen from the manuscript. But I have no insight/explanation in the causes of this performance. We considered removing this algorithm from the manuscript as it in any case is very slow for big datasets (also hindering too many re-analyses, etc.).

Line 203: This sentence ends abruptly. Where should we look to visualize this?

RESPONSE: sentence deleted (remnants of old text).

Line 205 (captions for Fig. 1f and Fig. S6): In the Taxonomic Dissimilarity section, the choice and interpretation of the dissimilarity metric could be explained more clearly. As written, it is hard to know what a “good” result would look like. Would a “perfect” result be taxonomic dissimilarity of zero for each plot? This gets a bit confusing, in part because the result is expressed as a before/after correlation and the eye has a tendency to want to see a strong correlation in Figs. 1f and S6. For this reason, I had to read through the text and captions several times to realize that the best result was not CROP. It is explained that the goal is to pull points to the “right” of the 1:1 line, but really the goal is to pull points close to 0 on the y-axis, regardless of where they start (left to right) along the x-axis. Once this is established, it does become clear that LULU improves the results of some algorithms without improving the results of others; nevertheless, there is variation among starting conditions based on the output of each algorithm (indeed, some do not require as much improvement, as indicated by their truncated distribution along the x-axis). Thus, while it is true that CROP is not improved as much as some of the others (e.g., SWARM), it is also true that CROP did not require as much improvement (at least according to this metric). What also appears true, is that a considerable amount of unexplained taxonomic dissimilarity remains following the application of LULU: far from having a taxonomic dissimilarity of 0, results range from ~0-100 across sites and algorithms. This very definitely must be acknowledged in the main text – it is promising that LULU shows improvement in some algorithms, but a more simple presentation (e.g., histogram) showing the final distribution of taxonomic dissimilarities would make it clear that these two datasets still do not correspond. (Again, whether or not they SHOULD perfectly correspond is another issue.)

RESPONSE: We agree, that these analyses were very difficult to interpret, and could be done more directly, as suggested in an earlier comment. Thus, as mentioned above, we have now replaced the section “Taxonomic dissimilarity” with a new section “Taxonomic composition” with more direct comparisons of the site-wise species/OTU composition. We hope this initiative is satisfactory.

Line 219: “Community composition is mainly driven” – not the composition of the community itself, just the metric that is calculated.

RESPONSE: Corrected. Section start now reads (L228):
“Metrics of community composition is mainly driven by...”

Lines 238-239: From the methods section it is clear that field-based matrices were not weighted by any measure of abundance. That should be made clear here as well, and it should be considered whether accounting for relative read abundances (vs. presence/absence) in the DNA matrix is sensible in this case.

RESPONSE: We have elaborated a little on the wording and included span and average of r-values before/after curation (L252-L254). As it is common practice to use abundance data for these metrics for sequence data, we find that some readers may find it valuable to be able to compare results from binary and abundance data, and thus we have retained both types of calculations. (L234-L238)

Lines 328: “alpha-diversity” – and “alpha-composition”? Both should be addressed.

RESPONSE: You are right. Based on the newly added section “taxonomic composition” we have added some words on “composition” as well in this sentence and some related places in the text.

Line 360: “adequate for studies of community dissimilarity” – but perhaps not for studies of community composition (i.e., the identity AND richness of species)?

RESPONSE: Of course (based on the results), it should be better for identity, composition and richness. Now reformulated. (L345,L369)

Lines 373-374: “CROP was the method least receptive...” – Again, is this because CROP started off in a “good” place, or is this lack of receptivity problematic in some way?

RESPONSE: It started in a “somewhat good place” - but the selected representative sequences for each OTU are suboptimal as they often have imperfect reference database matches. This has now been added to the text more clearly where relevant.

REVIEWERS' COMMENTS:

Reviewer #3 (Remarks to the Author):

The revisions by Frøslev et al. have improved the manuscript and I find it suitable for publication. The authors' conscientious and detailed responses—coupled with the inclusion of additional analyses—make the results much more transparent and leave me feeling more confident that the LULU algorithm is differentially retaining “correct” OTUs while minimizing the inadvertent loss of desirable OTUs. By acknowledging potential sources of mismatches between the field surveys and sequence data (besides sequencing error), and by quantifying the potential costs and benefits of applying the algorithm (both in terms of richness and composition), I think it will be much easier for readers to gauge how effective the algorithm may be currently as well as how future research may further improve it. Prior to publication, my only suggestion is to thoroughly copy edit the manuscript for grammatical errors and for opportunities to improve readability by simplifying sentence structure.

REVIEWERS' COMMENTS:

Reviewer #3 (Remarks to the Author):

The revisions by Frislev et al. have improved the manuscript and I find it suitable for publication. The authors' conscientious and detailed responses—coupled with the inclusion of additional analyses—make the results much more transparent and leave me feeling more confident that the LULU algorithm is differentially retaining “correct” OTUs while minimizing the inadvertent loss of desirable OTUs. By acknowledging potential sources of mismatches between the field surveys and sequence data (besides sequencing error), and by quantifying the potential costs and benefits of applying the algorithm (both in terms of richness and composition), I think it will be much easier for readers to gauge how effective the algorithm may be currently as well as how future research may further improve it. Prior to publication, my only suggestion is to thoroughly copy edit the manuscript for grammatical errors and for opportunities to improve readability by simplifying sentence structure.

RESPONSE: Thanks. We have tried our best to catch grammatical errors and improve the readability.